# The Advanced Infra-Red WAter Vapour Estimator (AIRWAVE) version 2: algorithm evolution, dataset description and performance improvements

Elisa Castelli[1], Enzo Papandrea[2,1], Alessio Di Roma[3], Bianca Maria Dinelli[1], Stefano Casadio[2], and Bojan Bojkov[4]

[1]Istituto di Scienze dell'Atmosfera e del Clima, ISAC-CNR, Via Gobetti 101, 40129 Bologna, Italy
[2]Serco s.p.a.,Via Sciadonna 24-26, 00044 Frascati, Italy
[3]Dipartimento di Fisica e Astronomia, DIFA, Universita' di Bologna, Viale Berti Pichat 6/2, 40127, Bologna, Italy
[4]EUMETSAT, Eumetsat Allee 1, D-64295 Darmstadt, Germany

*Correspondence to:* E. Castelli
(e.castelli@isac.cnr.it)

**Abstract.** Total Column Water Vapour (TCWV) is a key atmospheric variable which is generally evaluated at global scales through the use of satellite data. Recently a new algorithm, called AIRWAVE (Advanced Infra-Red Water Vapour Estimator), has been developed for the retrieval of the TCWV from the Along-Track Scanning Radiometer (ATSR) instrument series. The AIRWAVE algorithm retrieves TCWV exploiting the dual view of the ATSR instruments using the infra-red channels at 10.8 and 12 $\mu$m and nadir and forward observation geometries. The algorithm was used to produce a TCWV database over sea from the whole ATSR mission. When compared to independent TCWV products, AIRWAVE Version 1 (AIRWAVEv1) database shows very good agreement with an overall bias of 3% all over the ATSR missions. A large contribution to this bias comes from the polar and the coastal region where AIRWAVE underestimate the TCWV amount. In this paper we describe an updated version of the algorithm, specifically developed to reduce the bias in these regions. The AIRWAVE Version 2 (AIRWAVEv2) accounts for the atmospheric variability at different latitudes and the associated seasonality. In addition, the dependency of the retrieval parameters on satellite across-track viewing angles is now explicitly handled. With the new algorithm we produced a second version of the AIRWAVE dataset. As for AIRWAVEv1, the quality of AIRWAVEv2 dataset is assessed through the comparison with the Special Sensor Microwave/Imager (SSM/I) and with the Analyzed Radiosounding Archive (ARSA) TCWV data. Results show significant improvements in both biases (from 0.72 to 0.02 kg/m$^2$) and standard deviations (from 5.75 to 4.69 kg/m$^2$), especially in polar and costal regions. A qualitative and quantitative estimate of the main error sources affecting the AIRWAVEv2 TCWV dataset is also given. The new dataset has also been used to estimate the water vapour climatology from the 1991-2012 time series.

## 1 Introduction

A key issue in assessing the climate change is the precise knowledge of the distribution and variability of the Total Column of Water Vapour (TCWV), i.e. the vertically integrated atmospheric water vapour content. Actually, TCWV is closely linked

to clouds, precipitation and thus to the hydrological cycle (Allan et al., 2014). For this reason it is one of the GCOS (Global Climate Observing System) Essential Climate Variables (ECVs). Since water vapour plays such a crucial role in meteorological as well as in climatological aspects, it is important to gather spatial and temporal thorough information about its distribution. At global scale, this can be achieved through the use of satellite missions. In the last decades measurements from several sensors were used for this purpose. Among them, sensors operating in the microwave regions as the Special Sensor Microwave Imager (SSM/I onboard Defense Meteorological Satellite Program (DMSP) satellites) are used to infer accurate TCWV amount over ocean surfaces (Wentz, 1997), while sensors operating in the visible and near-infrared spectral range provide precise TCWV retrieval on land surfaces (e.g. Medium Resolution Imaging Spectrometer (MERIS)/ENVISAT (Lindstrot et al., 2012), or Moderate Resolution Imaging Spectroradiometer (MODIS) on the Terra and Aqua satellites (Diedrich et al., 2015)).

TCWV retrievals from infrared spectral regions were performed from Advanced Very High Resolution Radiometer (AVHRR; Emery, 1992) measurements, using the split window technique, (Sobrino et al., 1991; Li et al., 2003) and from MODIS (Seemann et al., 2003). TCWV retrievals from infrared channels over land suffer of the limited knowledge of the temperature and the emissivity of land surfaces (Lindstrot et al., 2014). The Along-Track Scanning Radiometer (ATSR, Delderfield et al. (1986)) instrument series had as main objective the accurate retrieval of sea surface temperature for climate studies. However, Casadio et al. (2016) demonstrated that it is possible to retrieve accurate and precise TCWV from its day and night time measurements, using the ATSR Brightness Temperature (BT) collected from nadir and forward views in the channels at 10.8 and 12 $\mu$m in clear sky day and night sea scenes. The algorithm (named AIRWAVE, Advanced Infra-Red Water Vapour Estimator) exploits a sea emissivity dataset and calculations performed with a dedicated Radiative Transfer Model (RTM). A detailed description of the AIRWAVE algorithm is given in (Casadio et al., 2016). The first version of AIRWAVE TCWV dataset (hereafter AIRWAVEv1), spanning from 1991 to 2012, is freely available from the GEWEX G-VAP website (G-VAP website) in form of monthly fields at $2° \times 2°$ regular grid resolution from 2003 to 2008 (Schroder et al., 2018). Due to the legacy of the ATSR series, and the fact that the radiances are a fundamental climate dataset record, the AIRWAVE dataset is an important resource for water vapor studies. It's worth underlying here that AIRWAVEv1 was developed to demonstrate the possibility of retrieving TCWV values from the ATSR measurements. The main goal pursued in its development was to have a simple software that could produce good results when compared to independent datasets. For this reason, in the AIRWAVEv1 algorithm several approximations were made. AIRWAVEv1 use fixed retrieval parameters along the globe and TCWV are corrected for viewing angles variability in nadir and slant by using an empirical correction factor.

Papandrea et al. (2018), aiming at the validation of the AIRWAVEv1 dataset, compared the data with the TCWV from SSM/I and Analyzed Radiosounding Archive, (ARSA website)) for the whole mission. This exercise demonstrated a general good quality of AIRWAVEv1 dataset (average correlative bias of 0.72 kg/m$^2$ vs SSM/I and 0.80 kg/m$^2$ vs ARSA, below the 1 kg/m$^2$ indicated in the GlobVapour project (Lindstrot et al., 2010)) apart for the polar regions and some coastal regions where an underestimation of the TCWV was found. In this paper we describe the new version of the AIRWAVE algorithm (hereafter AIRWAVEv2) developed to overcome these weaknesses by accounting for latitudinal and angular variations of the retrieval parameters. The new algorithm has been applied to all the available ATSR Level 1B Top of Atmosphere radiance products acquired over water surfaces in clear sky and in day/night conditions (same as for AIRWAVEv1) to produce the AIRWAVEv2

dataset. We show here the new TCWV climatologies derived from the 20 years of ATSR data together with the results of an extensive validation exercise performed repeating the same comparisons reported in (Papandrea et al., 2018). The new dataset shows improvements in terms of both bias and spread of the differences with respect to another dataset with respect to what achieved with AIRWAVEv1.

This article is structured as follows: In section 2 we describe the new algorithm developed to produce AIRWAVEv2, the improvements in retrieval scenarios and the strategy used for the selection of latitude and seasonal dependent retrieval parameters. In section 3 we describe the AIRWAVEv2 dataset, the TCWV climatology and its validation agains SSM/I and ARSA data and compare the performances of AIRWAVEv2 against AIRWAVEv1, finally, conclusions are given in section 4.

## 2   The AIRWAVE version 2

Papandrea et al. (2018) demonstrated the high quality of AIRWAVEv1 by comparing the retrieved TCWV with corresponding SSM/I and ARSA TCWV. However, in the same paper, the authors highlighted that at latitudes higher than 50° the agreement was not as good as for the rest of the globe. They speculated that this was due to the fact that AIRWAVEv1 makes use of retrieval parameters calculated though RTM simulations of tropical and mid-latitude atmospheric scenarios then averaged and used for the whole globe. This choice was driven by the consideration that, being AIRWAVEv1 applicable to water and

cloud-free scenes only, the number of cloud-free measurements over the sea at high latitudes is significantly smaller than at mid-latitudes and tropical regions. Thus, a trade-off between generality (i.e. good precision at all latitudes), actual latitudinal coverage of cloud-free measurements and software complexity was the main driver for this choice. Moreover, AIRWAVEv1 makes use of retrieval parameters computed for the along track viewing geometries only and uses an a-posteriori correction for the scenes pointing outside the orbit track.

The need to have a TCWV dataset of homogeneous quality at all latitudes and viewing geometries has driven the development of an improved version of the AIRWAVE algorithm, AIRWAVEv2. The improvements were achieved through three main steps. Firstly, we modified the way in which some of the approximations of the solving equations were handled, leading to an improved retrieval precision. Secondly, we compute the retrieval parameters for different latitude bands and for four months that, in the retrieval, are used as look-up-tables. Finally, we calculated the retrieval parameters for different viewing angles to

directly account for across track variations. We discuss these modifications in the following subsections. All the computations described in the paper have been made using the HITRAN2008 database (Rothman et al., 2009) for the spectroscopic data and the IG2 database (Remedios et al., 2007) version 4.1 for the atmospheric scenarios. The IG2 database was developed to be used as model atmosphere in the analysis of the Michelson Interferometer for Passive Atmospheric Sounding (MIPAS)/ENVISAT measurements (that cover the same spectral region of the IR channels of ATSR). The IG2 database contains atmospheric

vertical profiles of pressure, temperature and abundances of the molecules active in the the MIPAS spectral region, different for each year of the mission and divided into six latitudinal bands (polar, mid-latitude and equatorial for both North and South hemispheres) and four seasons.

## 2.1 Improvements in the solving equations

The starting point for the calculations of AIRWAVEv2 retrieval parameters is the master equation of AIRWAVE Version 1. Since the expressions are the same for both nadir and forward geometry, we report here the equations for the general case, omitting the subscripts NAD (for NADIR) of FWD (for FORWARD) geometries. We reproduce here some of the equations reported in (Casadio et al., 2016) to help the reader in the comprehension of this article. The master equation of AIRWAVE algorithm is eq. (12) of the above mentioned work (now eq. 1):

$$\ln \frac{J_1^{\lambda_1}}{J_2^{\lambda_2}} = \ln \frac{F_1^{\lambda_1}}{F_2^{\lambda_2}} + \ln \frac{\epsilon_1^{\lambda_1}}{\epsilon_2^{\lambda_2}} + \ln \frac{\gamma_1^{\lambda_1}}{\gamma_2^{\lambda_2}} + \lambda_2\tau_2 - \lambda_1\tau_1 \tag{1}$$

The subscripts 1 and 2 represents the terms calculated in the 10.8 and 12 $\mu$m channels respectively. $\lambda_1$ is the value of the frequency in the 10.8 $\mu$m channel and $\lambda_2$ in the 12 $\mu$m channel. $J_1$ is the radiance that reaches the TOA for the 10.8 $\mu$m channel, $J_2$ is the radiance that reaches the TOA for the 12 $\mu$m channel, F includes the atmospheric ($J_a$) and surface radiance ($J_s$) contribution and is F=1+$\frac{J_a}{e^{-\tau}J_s}$, $\epsilon$ is the sea emissivity, $\gamma$ is a constant arising from the Planck law, $\tau$ are the optical depths at the two wavelength. Since only $H_2O$ and $CO_2$ significantly affect the optical depth into ATSR Thermal Infrared (TIR) channels we can write:

$$\ln \frac{J_1^{\lambda_1}}{J_2^{\lambda_2}} = \ln \frac{F_1^{\lambda_1}}{F_2^{\lambda_2}} + \ln \frac{\epsilon_1^{\lambda_1}}{\epsilon_2^{\lambda_2}} + \ln \frac{\gamma_1^{\lambda_1}}{\gamma_2^{\lambda_2}} + \lambda_2\tau_2^{H_2O} - \lambda_1\tau_1^{H_2O} + \lambda_2\tau_2^{CO_2} - \lambda_1\tau_1^{CO_2} \tag{2}$$

then re-naming

$$G = \ln \frac{F_1^{\lambda_1}}{F_2^{\lambda_2}}, E = \ln \frac{\epsilon_1^{\lambda_1}}{\epsilon_2^{\lambda_2}}, \chi = \ln \frac{\gamma_1^{\lambda_1}}{\gamma_2^{\lambda_2}} \tag{3}$$

we get:

$$\ln \frac{J_1^{\lambda_1}}{J_2^{\lambda_2}} = G + E + \chi + \lambda_2\tau_2^{H_2O} - \lambda_1\tau_1^{H_2O} + \lambda_2\tau_2^{CO_2} - \lambda_1\tau_1^{CO_2} \tag{4}$$

In AIRWAVEv1 we assume that the optical depth ($\tau$) is the product of the vertical column of $H_2O$ by the relative effective absorption cross section ($\lambda\sigma$), normalised to the air mass factor (AMF) for the given line of sight angle:

$$\lambda_2\tau_2^{H_2O} - \lambda_1\tau_1^{H_2O} = \frac{\lambda_2\sigma_2 - \lambda_1\sigma_1}{AMF}TCWV \tag{5}$$

This equation shows that a linear behaviour exists between the water vapour optical depth and the TCWV. The linear dependence is exploited to solve the AIRWAVE equation and to retrieve TCWV.

In the development of AIRWAVEv2 we investigated the possibility to find a more accurate solution of the AIRWAVE equation still preserving the linear dependence between water vapour optical depth and TCWV. We recall here that in AIRWAVEv1

the water absorption cross sections were obtained using MODTRAN (Berk et al., 2008) while all the other values were obtained with the dedicated RTM, developed for ATSR measurements simulations and described in (Casadio et al., 2016). For AIRWAVEv2, a different approach was adopted for the calculation of effective absorption cross section. We have simulated ATSR synthetic radiances for the different atmospheric scenarios of the IG2 database and thus with different water vapor content. A detailed descriptions of these simulations is given in Sect. 2.2.

Using these simulations and ATSR-SSM/I collocated TCWV, we verified that $ln\frac{J_1^{\lambda_1}}{J_2^{\lambda_2}}$ correctly reproduces the real measurement behaviour as function of TCWV and that this relation is in first approximation linear. In Fig. 2 the colored dots represent the values of the logarithm of radiance ratio in equation 1 as a function of the TCWV for the different atmospheric scenarios. We report only the values obtained for the sub satellite scans using the IG2 water profiles for the Summer season multiplied for 0.5 and 1.5. The different colors represent different latitude bands (going from red for tropical to blue for polar). The grey dots represent the radiance ratio calculated from along track AATSR measurements on the 5 and 6 of August 2008 aggregated at SSMI resolution ($0.25° \times 0.25°$). The value of TCWV associated to each AATSR sub-satellite measurement was obtained from coincident SSMI measurements. In order to minimise the impact of random error, only measurements with SSMI pixel coverage (calculated as the ratio between the actual and the maximum number of ATSR measurements that can be present into a SSMI pixel) greater than 10% were used for this exercise. Figure 2 shows that: a) the simulated radiances correctly reproduce the real measurement behavior; b) the relation between the radiances and the TCWV can be considered as linear. Actually, we find in this case a correlation of 0.904 for real data and 0.92 for the simulated ones (p-value of $7.3 \times 10^{-05}$).

Therefore we can now re-write equation (4), isolating the terms that account for the water content ($\tau_1^{H_2O}$ and $\tau_2^{H_2O}$):

$$\ln \frac{J_1^{\lambda_1}}{J_2^{\lambda_2}} - G - \chi - E - \lambda_2 \tau_2^{CO_2} + \lambda_1 \tau_1^{CO_2} = \lambda_2 \tau_2^{H_2O} - \lambda_1 \tau_1^{H_2O} = \Delta\tau \tag{6}$$

The $G$ term of equations (3) and (4) is not as constant as supposed and partially verified in (Casadio et al., 2016), and may depend on the different water vapor content. For this reason, for each atmospheric scenario the average of all the $G$ values obtained with different water vapour content is used (to vary the water vapour content we multiplied the water vapor profile for 0.5, 0.75, 1., 1.25, 1.5). Equation (6) can thus be written as:

$$\ln \frac{J_1^{\lambda_1}}{J_2^{\lambda_2}} - G_{AVG} - \chi - E - \lambda_2 \tau_2^{CO_2} + \lambda_1 \tau_1^{CO_2} = \Delta\tau \tag{7}$$

Therefore for each scenario and geometry we can write:

$$\ln \frac{J_1^{\lambda_1}}{J_2^{\lambda_2}} - G_{AVG} - \chi - E - \lambda_2 \tau_2^{CO_2} + \lambda_1 \tau_1^{CO_2} = \Delta\tau = \Delta\sigma \cdot TCWV + \Delta\rho \tag{8}$$

In this equation, $\Delta\sigma$ and $\Delta\rho$ represent the slope and intercept of the straight line representing the behaviour of the term containing the radiances as a function of the TCWV. In the testing version of the AIRWAVEv2 code, we estimated these

parameters using the values of the radiances and the TCWV obtained perturbing the IG2 water vapor amount by a factor 0.5 and 1.5. Grouping the terms in equation (8) as in (Casadio et al., 2016) we get:

$$TCWV = \Phi - \frac{G}{\Delta\sigma} \tag{9}$$

where $\Phi$ is the "water vapor pseudo-column" that in AIRWAVEv2 is defined as :

$$\Phi = \frac{\ln \frac{J_1^{\lambda_1}}{J_2^{\lambda_2}} - \chi - E - \lambda_2 \tau_2^{CO_2} + \lambda_1 \tau_1^{CO_2} - \Delta\rho}{\Delta\sigma} \tag{10}$$

This formula is sightly different from the one used in (Casadio et al., 2016) due to the presence of the $\Delta\rho$ term:

$$\Phi = \frac{\ln \frac{J_1^{\lambda_1}}{J_2^{\lambda_2}} - \chi - E - \lambda_2 \tau_2^{CO_2} + \lambda_1 \tau_1^{CO_2}}{\Delta\sigma} \tag{11}$$

If in eq. 9 now we explicit the dependence on the viewing angles we get:

$$TCWV_{NAD} = \Phi_{NAD} - \frac{G_{NAD}}{\Delta\sigma_{NAD}} \quad and \quad TCWV_{FWD} = \Phi_{FWD} - \frac{G_{FWD}}{\Delta\sigma_{FWD}}$$

$$\Phi_{NAD} = \frac{\ln \frac{J_{1\,NAD}^{\lambda_1}}{J_{2\,NAD}^{\lambda_2}} - \chi - E_{NAD} - \lambda_2 \tau_{2\,NAD}^{CO_2} + \lambda_1 \tau_{1\,NAD}^{CO_2} - \Delta\rho_{NAD}}{\Delta\sigma_{NAD}} \quad with \quad \Phi_{FWD} = \frac{\ln \frac{J_{1\,FWD}^{\lambda_1}}{J_{2\,FWD}^{\lambda_2}} - \chi - E_{FWD} - \lambda_2 \tau_{2\,FWD}^{CO_2} + \lambda_1 \tau_{1\,FWD}^{CO_2} - \Delta\rho_{FWD}}{\Delta\sigma_{FWD}} \tag{12}$$

the TCWV can be estimated through the knowledge of $G$. Actually, for single view geometry the presence of the $G$ term can affect the accurate determination of TCWV. Thus, the $G$ variability suggests that it will be desirable to avoid this term in TCWV derivation. In AIRWAVE this is performed exploiting the dual view capability of the ATSR instruments and by assuming a perfect collocation between NAD and FWD measurements:

$$TCWV = \alpha \cdot \Phi_{NAD} + \beta \cdot \Phi_{FWD} \tag{13}$$

where

$$\alpha = \frac{1}{1 - \frac{\Delta\sigma_{FWD}}{\delta \cdot \Delta\sigma_{NAD}}} \quad and \quad \beta = \frac{1}{1 - \frac{\delta \cdot \Delta\sigma_{NAD}}{\Delta\sigma_{FWD}}}$$
$$with \quad \delta \approx \frac{G_{FWD}}{G_{NAD}} \tag{14}$$

Equations (10), (12) and (13) are the solving equations used in AIRWAVEv2, while AIRWAVEv1 makes use of equations (11), (12) and (13). Equations (10), (12) and (13) were solved for the 11 couples of viewing angles corresponding to the tie points. The angles cover a range from 0° to 21° in the NAD case and from 53° to 55° in the FWD case. The new equations were

used to compute a new set of retrieval parameters. For consistency purposes, in AIRWAVEv2 the computations were performed with the dedicated RTM as for AIRWAVEv1. An example of the difference between the parameters used for AIRWAVEv1 and AIRWAVEv2 is given in Table 1 for tropical scenario and sub-satellite view configuration. As can be noticed, the larger differences are for $\alpha$, $\beta$ and $\Delta\sigma$ parameters: there is a reduction of the $\alpha$ and $\beta$ parameters of a factor of about 50 from AIRWAVEv1 to AIRWAVEv2, while $\Delta\sigma$ is reduced by a factor of 30.

These changes have a direct effect on the retrieval precision. We can have an estimate of the improvements of the precision of AIRWAVEv2 retrievals against AIRWAVEv1 using the following consideration: in AIRWAVE we can estimate the expected precision by multiplying the measurement random error by a factor of $\alpha/\Delta\sigma$. Therefore in AIRWAVEv1 we multiply the random error by a factor of 53/1.9 (= 28) while in AIRWAVEv2 the multiplicative factor is 1.7/0.08 (= 21). In case of polar atmosphere this ratio is further reduced reaching a value of 13. Since the overall random error is about 0.25% for AATSR and

ATSR-2 and 0.6% for ATSR-1, we have that the precision for AATSR and ATSR-2 improves from 7% to 5% and for ATRS-1 goes from 17% to 12% for tropical atmosphere and in AIRWAVEv2 reaches 3% for AATSR ad ATSR-2 and 8 % for ATSR-1 in polar atmosphere. In worse cases, the precision has at maximum 1% higher value in the extreme across track of the swath with respect to the sub satellite points. From these considerations it follows that AIRWAVEv2 parameters should improve the retrieval performances.

**2.2   Improvements in the retrieval scenario**

In AIRWAVEv1 we make use of the same set of retrieval parameters for all measurements of a single ATSR instrument. Independent sets of parameters are calculated for the three missions, while within each mission no dependencies on different atmospheric/surface conditions or seasons was considered. In AIRWAVEv2 the retrieval parameters are estimated not only according to the instrument type but also accounting for possible latitudinal and seasonal variations. To this aim, we have used

the aforementioned RTM to compute all the required quantities exploiting the model atmospheres of the IG2 database. Since the IG2 database was specifically developed for the MIPAS/ENVISAT mission, it covers only the time range from 2002 to 2012, while the ATSR series operated from 1991 to 2012. However, the inter-annual variations of most of the species active in the ATSR thermal infrared spectral range are generally much smaller than the corresponding seasonal ones. Therefore we used the data for one year only (2010), and we considered the inter-annual variations as systematic error sources (see Appendix A

for an estimate of these errors).

To better reproduce the variability of the atmospheric scenarios that were observed by the ATSR instruments, we therefore calculated the retrieval parameters exploiting the profiles for all the six latitude bands and the four seasons included in the IG2 datasets for one year only.

The AIRWAVE solving equations do not explicitly make use of the Sea Surface Temperature (SST). However, it is used in

the Radiative Transfer (RT) computations to estimate the retrieval parameters. The sea surface emissivity values are instead used both in the parameters estimation and in the AIRWAVE retrieval. While over land the emissivity is characterised by a large spatial difference (it indeed varies as a function of soil type, vegetation cover, etc.), over sea its variation is in general relatively small. For this reason in AIRWAVEv1 we used constant emissivity values calculated for the nadir and forward viewing angles

with fixed SST (285K) and wind speed (3 m/s). In the new version of the algorithm, coherently with the approach used for the atmospheric scenarios, the retrieval parameters have been computed using dedicated SST values for each season and latitude band. The SST monthly means were produced for the corresponding six latitude bands and for the four seasons using ECMWF ERA-Interim daily fields data with a regular latitude/longitude grid of $0.75° \times 0.75°$ ($241 \times 480$ grid points).

The emissivity of each scenario has been computed using the data extracted from the University of Edinburgh database (Embury et al., 2008). This dataset contains emissivities tabulated as a function of wave number (600-3350 cm$^{-1}$ or 3-16.7 $\mu$m), viewing angle (0-85°), temperature (270-310 K), and wind speed (0-25 m/s at 12.5 m). For the RTM computations we used the full spectral dependency of the emissivity. Since in equation 3 of AIRWAVE we use a single emissivity value for each channel, we estimated it by convolving the spectral emissivity with the ATSR filter functions. The nadir and forward viewing angles of the instruments have been defined at eleven tie points of the ATSR swath (pixels associated with specific points equally spaced across a single image or instrument scan). For each tie point we then used the corresponding viewing angles to extract the correct emissivity values, with a fixed wind speed (3 m/s), as for AIRWAVEv1. Due to wind speed variability, in both algorithm version, we prefer to fix the value of the wind speed and then treat wind variations as an error source (see Sect. 5.3).

## 2.3 Across track variations of the retrieval parameters

A simplification present in AIRWAVEv1 is that the retrieval parameters are calculated only for the sub-satellite viewing angles (55° for forward view and 0° as nadir viewing angle). However, due to the ATSR configuration, the nadir viewing angles varies from 0° (sub-satellite) to approximately 21° (across track edge of the ATSR swath, $\pm$ 250 km from nadir), while the forward viewing angles range from 53° to 55°. Significant TCWV differences between centre and edge swath are, thus, expected.

In AIRWAVEv1 the across-track dependence of TCWV is corrected a-posteriori. The correction was calculated on the basis of TCWV retrievals performed over simulated brightness temperatures (BTs).

Figure 1 shows the absolute difference between TCWV from across track pixels and sub-satellite pixels calculated for AATSR in different atmospheric scenarios (tropical and mid-latitude) from synthetic measurements. To produce these retrievals, we simulated ATSR(s) radiances at the eleven tie points for the nadir and forward views considering also the surface emissivity variations with viewing angles. The atmospheric scenario, the TCWV and the SST were kept constant and they were exactly the same used for the sub-satellite track case. Then we retrieved the TCWV for each of the eleven couples of BTs and computed the difference with respect to the value obtained at the subsatellite track position (that coincide with the TCWV reference value). The dotted line in Fig. 1 mimics the correction term adopted within the AIRWAVEv1 algorithm. As can be seen the dotted line well reproduce the general behavior of the across track TCWV dependence. This is confirmed by the comparison made between AIRWAVEv1 TCWV and SSM/I or ECMWF TCWV (Casadio et al., 2016): on average we did not find any across-track bias, thus confirming the general validity of this correction.

However, as can be seen in Fig. 1, this approximation might not be sufficiently adequate depending on the used atmospheric scenario. Furthermore, a slight asymmetry with respect to the sub-satellite track position is expected as the ATSR instruments are tilted of about 4° respect to the flying direction of the satellite. Therefore the a-posteriori correction of AIRWAVEv1

cannot fully reproduce all these features. In AIRWAVEv2 we replaced the a-posteriori correction: We calculated the retrieval parameters for each of the above described tie points of the nadir and forward swaths, then we obtained the parameters at the
exact ground pixel position interpolating these values and using the ground pixel across track position.

## 2.4   Selection of the retrieval parameters

The computation of the retrieval parameters, described in the previous sections, produced a set of 1584 retrieval parameters for each ATSR mission (6 coefficients × 6 latitude bands × 4 seasons × 11 tie points) store in dedicated look-up tables (LUTs). In order to select the most suitable set of parameters for each ATSR measurement, a multivariate interpolation on a 3-dimensional
grid (trilinear) has been applied to the six retrieval coefficients ($\Delta\sigma$, $G$, $\Delta\rho$) and the emissivity for both the FWD and the NAD geometries.

## 3   AIRWAVEv2 dataset: description, climatology and validation

The AIRWAVEv2 TCWV data are produced, as for the AIRWAVEv1 dataset, processing Level 1B measurements acquired over water surfaces (sea and lakes) and in clear sky conditions in both nadir and forward views (according to the Level 1B cloud
mask). The output files are saved in Interactive Data Language (IDL) binary files (.sav extension), however they can be easily converted in other formats (e.g. netcdf) upon request. The parameters contained into the files are structured in two groups, in the first one (named HIRES) the parameters are given at native resolution (1 ×1 km$^2$) while in the second one (named SSM/I) the parameters have been aggregated to SSM/I resolution (0.25° × 0.25° grid). Both groups contain: the TCWV, the latitude, the longitude, the across track index value (0-512) and a day/night flag. The SSM/I group, in addition, contains the value of the
number of elements aggregated within the SSM/I grid cell and the standard deviation of the TCWV value associated to each cell.

The climatologies have been derived using all the available years and sensors of the ATSR family. Using the AIRWAVEv2 products aggregated at the 0.25° × 0.25° grid, we obtained for each month day/night TCWV averages and standard deviations. The final monthly files are also saved as IDL binary files but can be converted in other formats and are available upon request.
Figures 3 and 4 show sketches of the climatology for January, April, July and October obtained from 20 years of ATSR daytime measurements. Similar results are obtained for nigh-time retrievals (not shown here). In Fig.3 we report in panels a), c), e), g) the TCWV global distribution at 0.25° grid resolution and its standard deviation in panels b), d), f) and h). In Fig.4 panels a), c), e), g) we show the TCWV meridional mean together with its standard deviation, while on panels b), d), f) and h) we report the TCWV zonal mean.
The geographical distribution of the median values of the TCWV reflects the behaviour of general atmospheric circulation. Higher TCWV values, associated to strong convection in the Inter Tropical Convergence Zone (ITCZ), are located around the equator while lower values are found in the polar regions. Also the zonal means as a function of latitudes reflect this behaviour, while the zonal meridional means show a more homogeneous behaviour. Here lower TCWV values are found in coincidence of longitudes where we have extended land presence in the equatorial region. The mean and the absolute standard deviation show

similar features, with higher values in the ITCZ region. As can be noticed, the seasonal movements of the ITCZ from North (in Northern Hemisphere summer) to South (in Northern Hemisphere winter) can be clearly detected (Castelli et al., 2018). In general, the standard deviations in the region where the TCWV maximum is located are of the order of 15% and up to 20% in polar regions. The zonal means reflect the shift of the ITCZ during the year with maximum values of TCWV in Northern Hemisphere reached in July.

The quality of the AIRWAVEv2 dataset is evaluated through the same method adopted for AIRWAVEv1 and reported in (Papandrea et al., 2018). The AIRWAVEv2 dataset is compared to the TCWV obtained from the SSM/I satellite and to data available from the ARSA.

In this contest, these two datasets are complementary, as the SSM/I TCWV are not retrieved for measurements in proximity of coasts (minimum distance about 60 km), while the selected ARSA stations are located in coastal areas.

The SSM/I dataset is produced from the SSM/I instrument series onboard the DMSP polar orbit satellites since 1987. For this comparison, we used the 0.25° v7 daily product obtained from the F13 satellite produced by Remote Sensing Systems. In fact, differently from the other DMSP satellites, the local time of the ascending node of F13 (18:00 UTC) is more stable than the one of the other satellites with only a variation of 1 hour during the whole mission. For the comparison with SSM/I we used the AIRWAVEv2 data aggregated at 0.25° resolution covering the time period from 1995 to 2009. The ARSA dataset spans from January 1979 to present and contains water vapour concentration profiles at specific pressure levels. To obtain the TCWV we vertically integrate these values. For the comparison with the ATSR data, only stations surrounded, even partially, by water are used. More details about the selection of ATSR and ARSA coincident data are reported in (Papandrea et al., 2018).

The zonal means (calculated in bins of 2 degrees for all the datasets, and reported in Fig. 5) show the good quality of AIRWAVEv2 data with respect to both radiosondes and satellite data at all latitudes. In Fig. 5, the AIRWAVEv1 data are overplotted for comparison. The improvement in the performances of the new dataset is clearly visible at all latitudes, and in particular for regions with latitude higher than 45-50° where the negative values obtained with AIRWAVEv1 disappear. In addition, a significant reduction of the spread is highlighted.

In Fig. 6 we show the Bi-dimensional histograms of the comparisons between AIRWAVEv2 and SSM/I (panel a) and ARSA (panel b). The SSM/I measurements are homogeneously distributed over the globe, while ARSA radiosounding stations are mainly located at mid-latitudes (see Fig.7) and this is reflected in the bulk of the ARSA TCWV values ranging between 0-30 kg/m$^2$.

Globally, a good correlation is obtained against both datasets, as highlighted by the correlations (0.948 with SSM/I and 0.918 with ARSA) and bias values ($0.02 \pm 4.79$ kg/m$^2$ with respect to SSM/I and $0.19 \pm 6.12$ kg/m$^2$ with respect to ARSA). We highlight that, in the validation exercise, we compare SSM/I data in coincidence with AIRWAVEv2 ones. Since AIRWAVE is applicable only to clear sky measurements this is a method to filter out SSM/I cloudy TCWV and thus to avoid biases due to different sensitivity related to the used spectral range. When comparing to radiosondes, the small bias we found demonstrate that AIRWAVE TCWV are sensitive also to low atmospheric levels. In particular the comparison with the same histograms of AIRWAVEv1 (see Fig. 1 of Papandrea et al. (2018)) highlights the correction of negative values in polar and costal regions in the new version of the dataset.

Figure 7 reports the geographical distribution of the mean TCWV differences with respect to SSM/I and ARSA in absolute and percentage values. In comparison to AIRWAVEv1, see Fig. 3 of Papandrea et al. (2018), the differences with SSM/I are reduced at all latitudes. The longitudinal patterns of the differences are similar to the ones of AIRWAVEv1 except for the equatorial pacific region where AIRWAVEv2 shows a slightly higher positive bias. The reasons for this behaviour are under investigation. In the majority of costal regions, where AIRWAVEv1 underestimated the TCWV, AIRWAVEv2 is now in agreement with ARSA results (no SSM/I data close to coast).

In Table 2 we summarise the results of this comparison for both ARSA and SSM/I and for different scenarios and missions. The average bias is about $0.0 \pm 4.7$ kg/m$^2$ with respect to SSM/I and $0.2 \pm 6.1$ kg/m$^2$ with respect to ARSA. If we compare these results with the ones for AIRWAVEv1 (reported in Table 2 to ease the comparison) we can clearly see the improvement in both the biases and the standard deviations($0.7 \pm 5.7$ kg/m$^2$ with respect to SSM/I and $0.8 \pm 7.7$ kg/m$^2$ with respect to ARSA). As can be seen from Fig.s 5 and 8 and from results in Table 2 the improvement in the bias is obtained at all latitudes; it is however more evident in polar regions (from $5.5 \pm 5.1$ kg/m$^2$ in AIRWAVEv1 to $1.3 \pm 3.5$ kg/m$^2$ in AIRWAVEv2 versus SSM/I and from $4.1 \pm 6.5$ kg/m$^2$ in AIRWAVEv1 to $0.9 \pm 4.6$ kg/m$^2$ in AIRWAVEv2 when using ARSA). Slight difference between the three ATSR missions are consistent with the related uncertainties.

In Fig. 8 we show monthly mean evolution of the differences (and their standard deviation), between the TCWV obtained from correlative measurements and AIRWAVEv2. As for AIRWAVEv1, these differences are quite stable over time, with the exception of the beginning of the ATSR-1 mission (1991-1994). As explained in Papandrea et al. (2018), this can be due to the failure of 3.7 $\mu$m channel that impacted the ATSR cloud screening. In general, the differences with respect to the radiosonde exhibit a higher seasonality due to pronounced variability of atmospheric and surface conditions in costal areas.

It is worth noticing that the spread of the AIRWAVEv2 is always smaller than the one of AIRWAVEv1. This is partially due to the new algorithm that reduces the random error component due to noise on the retrieved TCWV.

The above described results indicate that the AIRWAVEv2 algorithm reduces the global bias with respect to SSM/I, from about 0.7 kg/m$^2$ of AIRWAVEv1 to 0.0 kg/m$^2$ and improves the standard deviations (STD) of up to 20% with respect to AIRWAVEv1. In AIRWAVEv2, the STD values are essentially constant for all the scenarios and missions, highlighting the un-biased nature of the dataset with respect to SSM/I (cloud-free). When using ARSA data the bias reduces from 0.8 kg/m$^2$ to 0.2 kg/m$^2$ and the standard deviations of 21%.

## 4 Discussion and Conclusions

The second version of the AIRWAVE TCWV dataset described in this work, has been validated against ARSA and SSM/I equivalent products.

As expected also from the analysis of synthetic retrievals, the most significant AIRWAVEv2 improvement is achieved at polar latitudes. In polar regions the bias versus SSM/I improves of 4.2 kg/m$^2$ and of 3.2 kg/m$^2$ versus ARSA. In both cases the standard deviations are reduced of about 1.6-1.9 kg/m$^2$ However, improvements at mid-latitudes are also found. The average bias with respect to SSM/I improves of about 0.7 kg/m$^2$ and the standard deviation is reduced of about 1 kg/m$^2$. In case of

validation against radiosondes the bias in AIRWAVEv2 is reduced of about 0.6 kg/m$^2$ with respect to AIRWAVEv1 and the standard deviation is reduced of 1.6 kg/m$^2$.

These improvements are due to two factors. First the AIRWAVEv2 retrieval parameters now account for the atmospheric variability. Secondly the implementation of the new algorithm explicitly takes into account the geometry and latitude dependence of each pixel, allowing to overcome possible artefacts due to approximations and a posteriori corrections. No statistically significant trend can be found in the comparison with SSM/I and ARSA in both versions of the database, while a seasonal dependence of the differences is observed, with larger bias in July and August, mainly due to the differences in mid-latitude

north TCWV retrievals. In general, we find slightly drier results with respect to ARSA and SSM/I with both versions. This is possibly due to the fact that the temporal mismatch between the ATSR and the correlative measurements does not allow to exclude all SSM/I TCWV retrieval obtained under cloudy conditions, or to wrong cloud mask assignation. As discussed, the use of retrieval parameters that are calculated in conditions different from the ones present in the observed scenario can cause biases on the obtained TCWV. We point out that the major source of errors on the retrieved TCWV comes from the

temperature profile assumptions, while erroneous assumptions of other gases (e.g. $HNO_3$, CFC-11, CFC-12, $CO_2$) have an almost negligible impact. The obtained RMSE value of about 7% is of the same order of this error. The error quantification given in this work allows the users to get a better insight of the AIRWAVEv2 TCWV dataset and of its related quality.

    Beside the improvements on ATSR TCWV retrievals given by the AIRWAVE Version 2 dataset, the method described in this work can be the basis for a similar approach for SLSTR (Sea and Land Surface Temperature Radiometer, on board

COPERNICUS SENTINEL-3, (Donlon et al., 2012)).

**Acknowledgements**

This work has been performed under the ESA-ESRIN Contract No. 4000108531/13/I-NB. The authors gratefully acknowledge ECMWF for SST data, the GlobVapour project for providing the combined MERIS+SSM/I water vapour product. SSM/I and SSM/IS data are produced by Remote Sensing Systems. Data are available at www.remss.com/missions/ssmi.

**5    Appendix A: Evaluation of AIRWAVE Version 2 main systematic and random errors**

This appendix provides an estimate of the main error sources (random and systematic) that affect the AIRWAVEv2 dataset. To facilitate the use of the errors together with the TCWV contained into the dataset, we summarize the results of this appendix in Table 3. AIRWAVEv2 retrieval parameters accounts for atmospheric and surface variability of the observed scenes. However, real observed scenes can obviously deviate from the simulated ones. An evaluation of the errors induced by these deviations

is then required. For this reason we analyse, through the use of synthetic radiances, the major sources of errors that can affect AIRWAVEv2 retrieval i.e. (in order of importance): the influence of the atmospheric temperature profile variations, the impact of sea emissivity changes due to the wind and the impact of interfering atmospheric species. Furthermore, an estimate of the random error component due to the noise is also reported.

## 5.1 Retrieval approximations

In order to evaluate the impact of systematic errors due to the retrieval approximations, we performed some tests of TCWV retrievals from simulated radiances. We used Top Of the Atmosphere (TOA) sub-satellite track Brightness Temperatures (BTs) simulated with the RTM at 10.8 and 12 $\mu$m in nadir and forward geometries as input to the retrieval chain (no random noise was added). The BTs were produced using different TCWV amounts for each given scenario (e.g. water vapour profiles were multiplied by 0.5, 0.75, 1.25, 1.5), while all the other atmospheric profiles were kept constant. Results of this exercise show that TCWV is correctly retrieved with an error of $\pm$ 3%. For comparison purposes, we performed a similar test using the AIRWAVEv1 approach. For AIRWAVEv2 tests, the same atmospheric conditions used to compute the retrieval parameters have been adopted, while for AIRWAVEv1 larger errors are expected due to the fact that the atmospheric variability is not taken into account. Despite this, this analysis shows that AIRWAVEv1 performs well for medium-high TCWV ($>$30 kg/m$^2$), where the differences are below 10%. For TCWV values between 10-20 kg/m$^2$, the differences range between -10 and -30%. AIRWAVEv1 underestimates the low TCWV values (below 10 kg/m$^2$) with differences going from -50% to -150/-200%. This was also reported in (Casadio et al., 2016), where the authors compared AIRWAVE TCWV with collocated ECMWF counterparts showing a dry bias at high latitudes (where the TCWV values are small). The tests on synthetic radiances indicate that the AIRWAVEv2 parametrisation solves this issue.

## 5.2 Atmospheric temperature and water vapor profiles and SSTs

One of the main error sources that can affect the AIRWAVE TCWV retrieval is the assumption of a fixed temperature profile. Actually, the atmospheric opacity is closely linked to the atmospheric density and thus to the atmospheric temperature. To estimate the impact of temperature on the retrieved TCWV, twenty different temperature profiles were randomly perturbed by $\pm$3 K on a 1 km equi-spaced altitude grid. In order to account also for possible changes related to SST variation we change the SST accordingly to the value of the temperature in the lowest layer. Then, simulated BTs were produced with the RTM and were used to perform the TCWV retrievals for the three instruments in equatorial, mid-latitude and polar July conditions for the North Hemisphere and the results were compared with respect to the unperturbed case. In the third column of Table 3 we summarise the findings of this analysis for each of the three instruments, reporting the STD of the difference both in absolute and in percentage values. The impact of these perturbations is of the order of 6% and is higher in the equatorial and mid-latitude regions and lower at the poles. Indeed, in the equatorial region, due to the higher water vapour content, the atmosphere is more opaque than at the poles so that temperature variations have a larger impact on the retrieved TCWV. These tests were also performed varying the atmospheric profile alone. Very similar but slightly higher errors are found in these cases. Another relevant error sources can be due to differences in water vapor profile shape. To evaluate this error, we varied the water vapor profile randomly up to 5% at each atmospheric level. At maximum the impact is of 1% in equatorial case (atmospheric opacity, see the fourth column in Table 3).

## 5.3 Wind Speed

A further source of error that can affect the TCWV retrievals is the value used for the sea emissivity, which enter directly into equations (10) and (11). Sea emissivity depends upon wavelenght, sea surface temperature, viewing angles and wind speed. As stated in Sect. 2.2, in AIRWAVEv2 we accounted for emissivity variations due to the viewing angles and sea surface temperatures. All the calculations were performed at a fixed wind value of 3 m/s. In order to assess the possible systematic effects due to wind variations on the retrieved values, we varied the emissivity according to the wind speed at three values as tabulated in the University of Edinburgh emissivity database: 1 m/s, 10 m/s and 25 m/s.

The emissivity has a different value and spectral behaviour with different wind speeds. The fourth column of Table 3 report the difference of the TCWV retrieved for simulated measurements with the 25 m/s wind speed with respect to the reference case. As expected, the impact is almost negligible in the equatorial band, where the higher opacity of the atmosphere reduces the sensitivity of ATSR measurements to the surface conditions, while increases toward the poles where, due to the low atmospheric opacity, the surface effects become relevant with respect to the atmospheric component. Furthermore, in case of polar conditions, the effects of the wind on the retrieved TCWV is not linear, with enhanced variations for wind speed of 25 m/s, as not only the intensity but also the spectral shape of the surface emissivity varies in function of the wind speed. The possible presence of white caps, generated by high speed winds, has not been considered in this study

## 5.4 Interfering atmospheric constituents

The AIRWAVE algorithm, in both versions, accounts for the contribution to the radiance of the two main gases active into the ATSR channels ($H_2O$ and $CO_2$). However some other species have spectroscopic features in the ATSR channels. In Fig. 9, the $CO_2$, $HNO_3$ and CFCs spectra in the 10-13 $\mu$m wavelength range are shown, along with the ATSR filter functions (all in arbitrary units). In order to have a complete view of the possible error components, we assessed the interfering species impact in case their abundance differs from the one used in the reference scenarios to compute the retrieval parameters.

Among the considered species, $HNO_3$ shows significant latitudinal and seasonal variability while $CO_2$ and CFCs exhibit inter-annual trends (Remedios et al., 2007). For this reason we separately accounted for the effects due to latitudinal and inter-annual variability. We used the IG2 database version 4.1 and our RTM to produce synthetic BTs. For each season/latitude band we generated synthetic BTs using the proper IG2 atmospheric status but the profile of the investigated species, for which we used all the different available profiles. The generated BTs were then used to retrieve the TCWV to assess the systematic error induced by the expected variability of the interfering species.

Latitudinal and seasonal variations of $HNO_3$ impact ATSR-1 and ATSR-2 BTs more than AATSR, because of the different shapes of the TIR filter functions, and can produce differences up to 0.3 K in the 11 $\mu$m band (mid-latitude vs tropical north in January). For CFC-11 seasonal differences in the tropics are of the order of 0.001 K in the 12 $\mu$m band and for CFC-12 we get 0.03 K in the 11 $\mu$m band, while latitudinal variations reach 0.04 K from tropical to mid-latitude atmospheres for both channels. $CO_2$ latitudinal variations can produce a maximum difference of 0.003 K on nadir BT. The impact of maximum latitudinal variations of $HNO_3$, CFCs and $CO_2$ on the retrieved TCWV is reported in Table 3. The largest contribution is due

to $HNO_3$ latitudinal variation and is of the order of 0.6%, while the CFCs latitudinal variations produce differences of 0.15% and CO2 of 0.01% only. Furthermore, the impact of using mid latitude profiles instead of tropical profiles for all species but $H_2O$ has a maximum impact of 0.22%. Seasonal variations are almost negligible for $CO_2$ and CFCs, while they are only 0.6% for $HNO_3$. Thus we can safely assume that the latitudinal and seasonal variations of the interfering species represent a minor error source for the AIRWAVE TCWV for both AIRWAVEv1 and AIRWAVEv2.

To evaluate the impact of inter-annual variations also for all the ATSR series, we would need the $CO_2$ and CFCs profiles from 1991 to 2012. However, as mentioned in the previous sections, the IG2 database contains data from the 2002 onwards. In this work, the CFCs and $CO_2$ profiles from 1991 to 2001 were inferred scaling the 2002 profiles using the trend given in the last IPCC report for $CO_2$ and in the Mauna Loa observatory website ((Global Monitoring Division website) website, see also Aoki et al. (2003) and Minschwaner et al. (2013)) for CFC-11 and CFC-12. The impact on the ATSR BTs due to the inter annual variations of $CO_2$ and CFCs has been evaluated for each mission. We calculated the synthetic spectra using the CFC-11, CFC-12 or $CO_2$ profile for the initial and final year of each mission. Then we calculated, for each instrument, the differences of retrieved TCWV at the beginning and at the end of each mission. Results for $CO_2$ and CFCs are shown in Table 3. The influence of $CO_2$ annual variations on the retrieved TCWV is of the order of 0.004 kg/m$^2$ ( <0.01 %). For CFCs we obtain 0.002 kg/m$^2$. We can then conclude that the impact of VMR latitudinal and seasonal variations on retrieved TCWV is negligible (maximum value 0.6%), and that the systematic effect of $CO_2$ and CFCs long term variations over the missions are even smaller (0.07%).

## 5.5 Noise

Finally we analyse the impact of the measurement noise on the retrieved TCWV (see last column of Table 3). The measurement noise was simulated applying a random perturbation of $\pm 0.037$ K on the BTs of the two channels of ATSR-2 and AATSR and a perturbation of $\pm 0.1$ K on the ATSR-1 channels (Smith et al., 2012). For each instrument, we have generated one thousand values and we have evaluated the standard deviation of the obtained TCWV that we report in absolute and percentage values in Table 3. The standard deviation is maximum at the poles, as expected since there the TCWV and S/N ratio are lower. For AIRWAVEv2 we get 18% for ATSR-1 and 6% for ATSR-2 and AATSR in the worst case. To be noticed that AIRWAVEv2 approach has, for all the scenarios, better performances with respect of AIRWAVEv1 (see Casadio et al. (2016)).

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

**Table 1.** AIRWAVE v1 and AIRWAVEv2 retrieval parameters for tropical summer atmosphere and along track configuration (sub-satellite view at $0°$ and $55°$).

| | | $\alpha$ | $\beta$ | $\delta$ | $\Delta\sigma_{NAD}$ $10^{-06}$ | $\Delta\sigma_{FWD}$ $10^{-06}$ | $G_{NAD}$ $10^{-06}$ | $G_{FWD}$ $10^{-06}$ | $\Delta\rho_{NAD}$ $10^{-06}$ | $\Delta\rho_{FWD}$ $10^{-06}$ |
|---|---|---|---|---|---|---|---|---|---|---|
| ATSR-1 | AIRWAVEv1 | 50.7 | -49.7 | 1.65 | 1.49 | 2.41 | -5.30 | -8.82 | — | — |
| | AIRWAVEv2 | 1.72 | -0.72 | 1.69 | 0.06 | 0.05 | -6.59 | -11.1 | 6.59 | 11. 2 |
| ATSR-2 | AIRWAVEv1 | 50.5 | -49.5 | 1.63 | 1.74 | 2.78 | -6.36 | -10.4 | — | — |
| | AIRWAVEv2 | 1.65 | -0.65 | 1.67 | 0.07 | 0.04 | -7.88 | -13.1 | 7.89 | 13.2 |
| AATSR | AIRWAVEv1 | 53.1 | -52.1 | 1.62 | 1.90 | 3.02 | -7.06 | -11.5 | — | — |
| | AIRWAVEv2 | 1.67 | -0.67 | 1.66 | 0.08 | 0.05 | -8.74 | -14.5 | 8.77 | 14.6 |

**Table 2.** AIRWAVE TCWVs compared with SSM/I and ARSA stations. Results are given also for AIRWAVEv1. Absolute differences along with the standard deviations are reported for the global (all latitudes), equatorial ($25°$ S-$25°$ N), mid-latitude (25-60$°$ S and 25-60$°$ N) and polar (>60$°$ N or >60$°$ S) scenarios. Average values for the ATSR-1, ATSR-2 and AATSR are also provided.

| Instrument | Scenario | N $\times10^5$ | BIAS-v2 kg/m$^2$ | STD-v2 kg/m$^2$ | BIAS-v1 | STD-v1 | N $\times10^5$ | BIAS-v2 kg/m$^2$ | STD-v2 kg/m$^2$ | BIAS-v1 | STD-v1 |
|---|---|---|---|---|---|---|---|---|---|---|---|
| | | | SSM/I-AIRWAVE | | | | | ARSA-AIRWAVE | | | |
| All | Global | 3110 | 0.02 | 4.69 | 0.72 | 5.75 | 3.01 | 0.19 | 6.12 | 0.80 | 7.73 |
| All | Equator | 1560 | -0.17 | 4.79 | -0.17 | 5.57 | 0.87 | -0.70 | 6.60 | -2.40 | 7.74 |
| All | Midlat | 1380 | 0.07 | 4.84 | 1.12 | 5.89 | 1.80 | 0.49 | 6.10 | 1.69 | 7.44 |
| All | Polar | 170 | 1.32 | 3.51 | 5.55 | 5.14 | 0.35 | 0.86 | 4.59 | 4.12 | 6.47 |
| ATSR-1 | Global | 190 | -0.20 | 5.17 | 1.15 | 6.17 | 0.48 | -0.71 | 6.24 | 0.23 | 7.62 |
| ATSR-2 | Global | 1390 | 0.24 | 4.77 | 0.80 | 5.87 | 1.00 | 0.70 | 6.04 | 1.13 | 7.65 |
| AATSR | Global | 1520 | -0.16 | 4.70 | 0.58 | 5.77 | 1.53 | 0.13 | 6.11 | 0.75 | 7.81 |

**Table 3.** Impact of different error sources on AIRWAVEv2 dataset.

| Instrument | Scenario | Temp profile+SST $kg/m^2$ | $H_2O$ profile $kg/m^2$ | wind (25 m/s) $kg/m^2$ | $HNO_3$ $kg/m^2$ | CFC-11 $kg/m^2$ | CFC-12 $kg/m^2$ | $CO_2$ $kg/m^2$ | CFC-11 trends $kg/m^2$ | CFC-12 trends $kg/m^2$ | $CO_2$ trends $kg/m^2$ | Noise $kg/m^2$ |
|---|---|---|---|---|---|---|---|---|---|---|---|---|
| ATSR-1 | Equatorial | 3.2 (6.1%) | 0.5 (1.0%) | -0.003 (-0.01%) | 0.2 (0.4%) | 0.07 (0.14%) | -0.07 (-0.16%) | 0.003 (0.006%) | -0.006 (-0.01%) | 0.014 (0.03%) | 0.01 (0.03%) | 4.8 (9%) |
|  | Midlat | 1.2 (4.2%) | 0.1 (0.5%) | 0.214 (0.69%) |  |  |  |  |  |  |  | 4.8 (16%) |
|  | Polar | 0.3 (1.6%) | 0.1 (0.5%) | 0.448 (2.70%) |  |  |  |  |  |  |  | 3.0 (18%) |
| ATSR-2 | Equatorial | 3.5 (6.6%) | 0.5 (1.0%) | -0.008 (-0.01%) | 0.15 (0.3%) | 0.05 (0.1%) | -0.05 (-0.1%) | -0.003 (-0.006%) | 0.015 (0.03%) | 0.009 (0.02%) | -0.02 (-0.05%) | 1.6 (3.1%) |
|  | Midlat | 1.4 (5.1%) | 0.2 (0.6%) | 0.230 (0.74%) |  |  |  |  |  |  |  | 1.6 (5.2%) |
|  | Polar | 0.4 (2.0%) | 0.1 (0.5%) | 0.467 (2.8%) |  |  |  |  |  |  |  | 1.0 (6.1%) |
| AATSR | Equatorial | 3.5 (6.7%) | 0.5 (1.0%) | -0.012 (-0.02%) | 0.28 (0.56%) | 0.08 (0.17%) | -0.07 (-0.15%) | 0.006 (0.01%) | 0.01 (0.02%) | 0.01 (0.02%) | -0.03 (-0.07%) | 1.4 (2.8%) |
|  | Midlat | 1.5 (5.4%) | 0.2 (0.6%) | 0.204 (0.66%) |  |  |  |  |  |  |  | 1.5 (4.8%) |
|  | Polar | 0.4 (2.2%) | 0.1 (0.5%) | 0.447 (2.7%) |  |  |  |  |  |  |  | 0.9 (5.6%) |

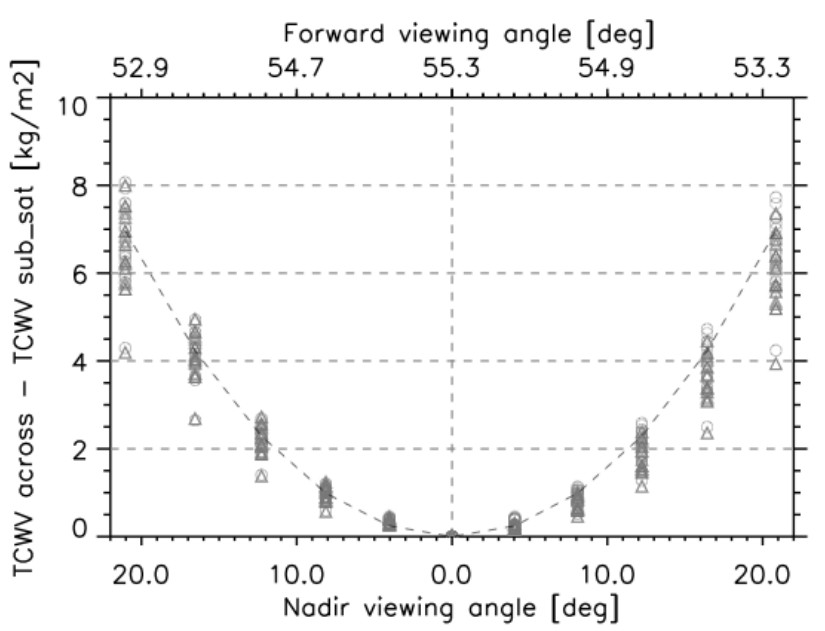

**Figure 1.** Differences between TCWV calculated from across pixels and TCWV calculated sub-satellite track as a function of across track position calculated for different atmospheric scenarios and for AATSR (symbols). Grey dashed line corresponds to AIRWAVEv1 parametrization.

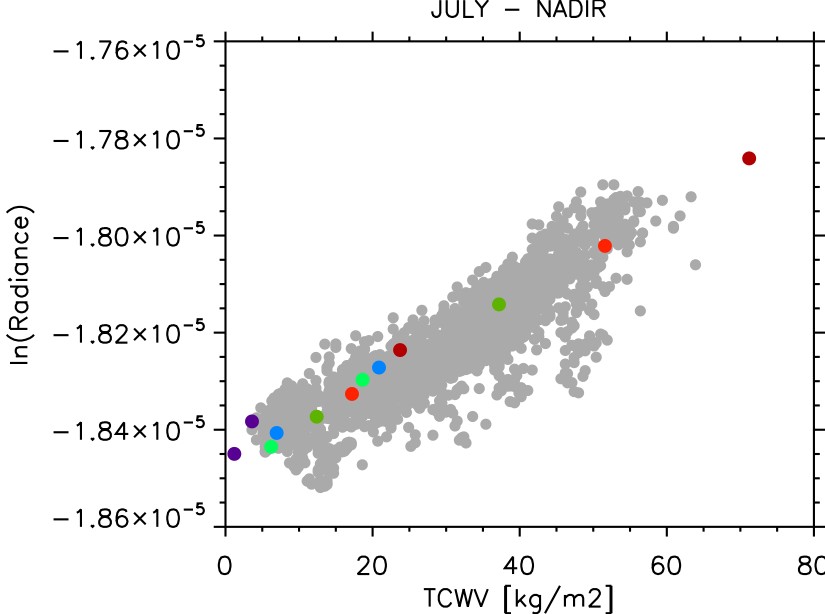

**Figure 2.** Logarithm of radiance ratio at nadir as a function of TCWV in simulated atmospheric scenarios (purple and blue for polar, light and dark green for mid-latitude and light and dark red for equatorial). Grey dots represent the same quantity using real AATSR radiances and coincident SSM/I TCWV for the along track measurements on the 5 and 6 August 2008.

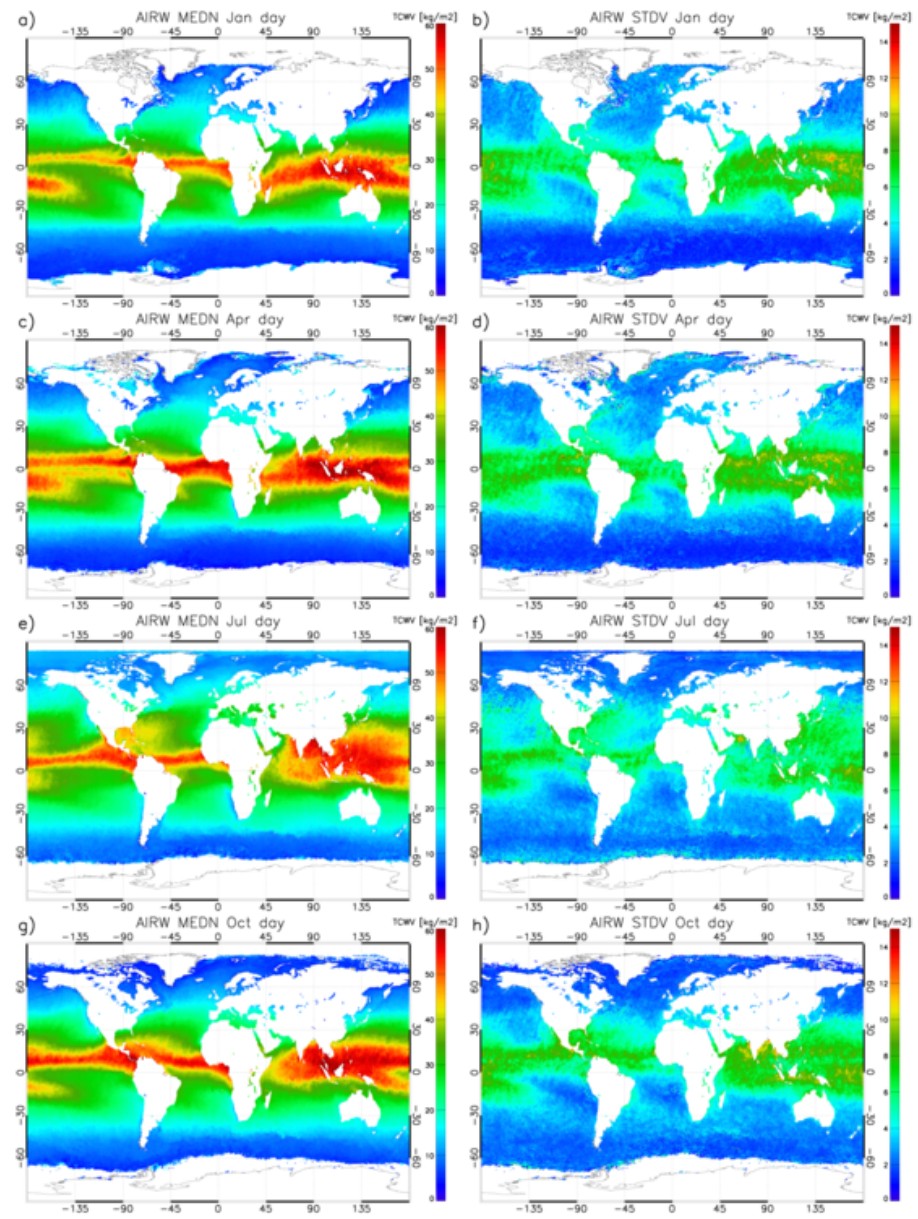

**Figure 3.** Climatology of daytime TCWV from AIRWAVEv2 dataset for January (a), April (c), July (e) and October (g) from 1991-2012, and standard deviations for same moths (respectively b, d, f, h).

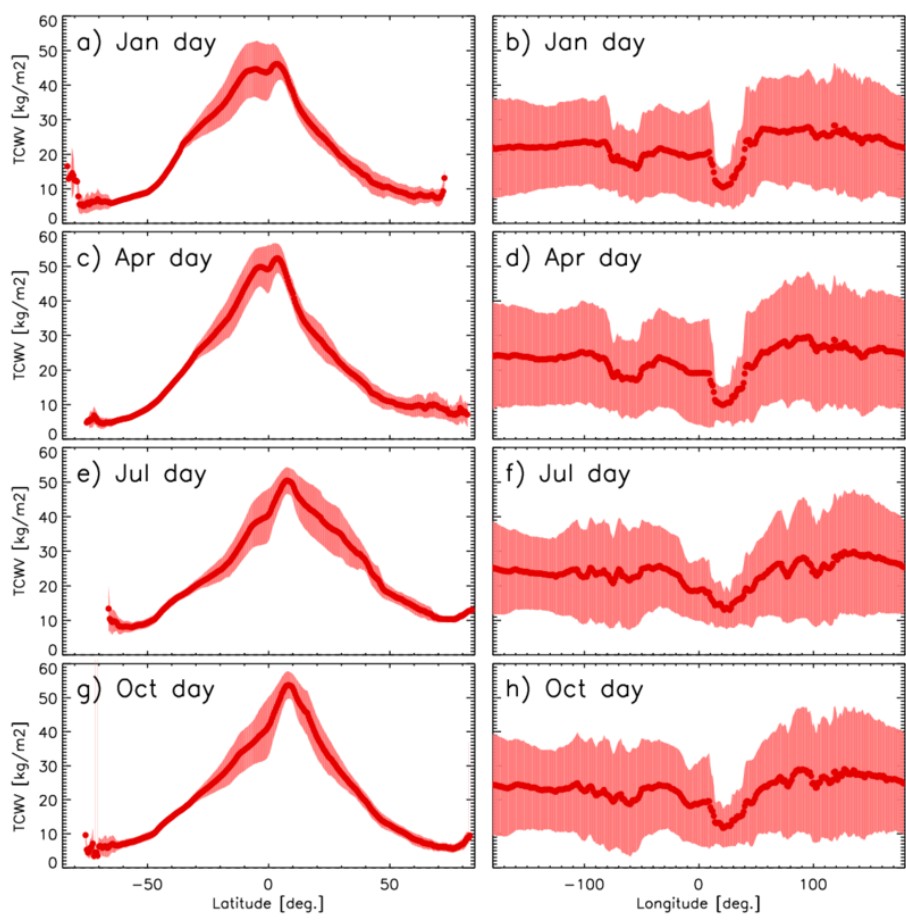

**Figure 4.** TCWV distribution as function of latitude (median and standard deviation) (a, c, e, g) and as function of longitude (b, d, f, h).

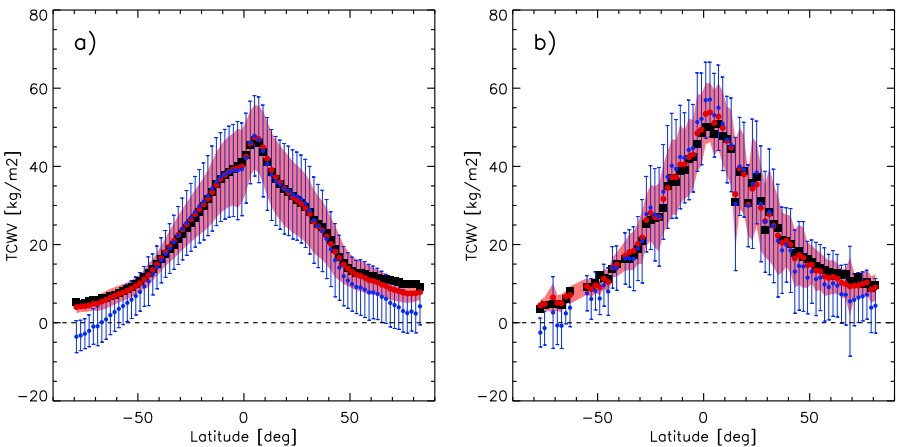

**Figure 5.** Zonal means of TCWV for AIRWAVEv1 (blue), AIRWAVEv2 (red) and correlative measurements (black): SSM/I (a) or ARSA (b). The data have been averaged in 2 degrees latitude bins. ARWAVE TCWVs standard deviations are also reported.

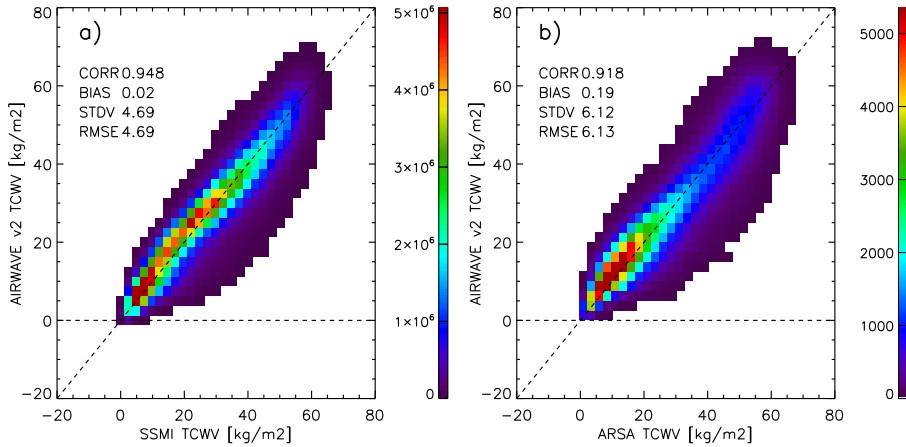

**Figure 6.** AIRWAVE TCWV vs SSM/I TCWV (a) or ARSA TCWV (b). The bin size is 2.5 kg/m$^2$. The colour scale indicates the number of elements of the histogram. The data cover the period from 1991 to 2012.

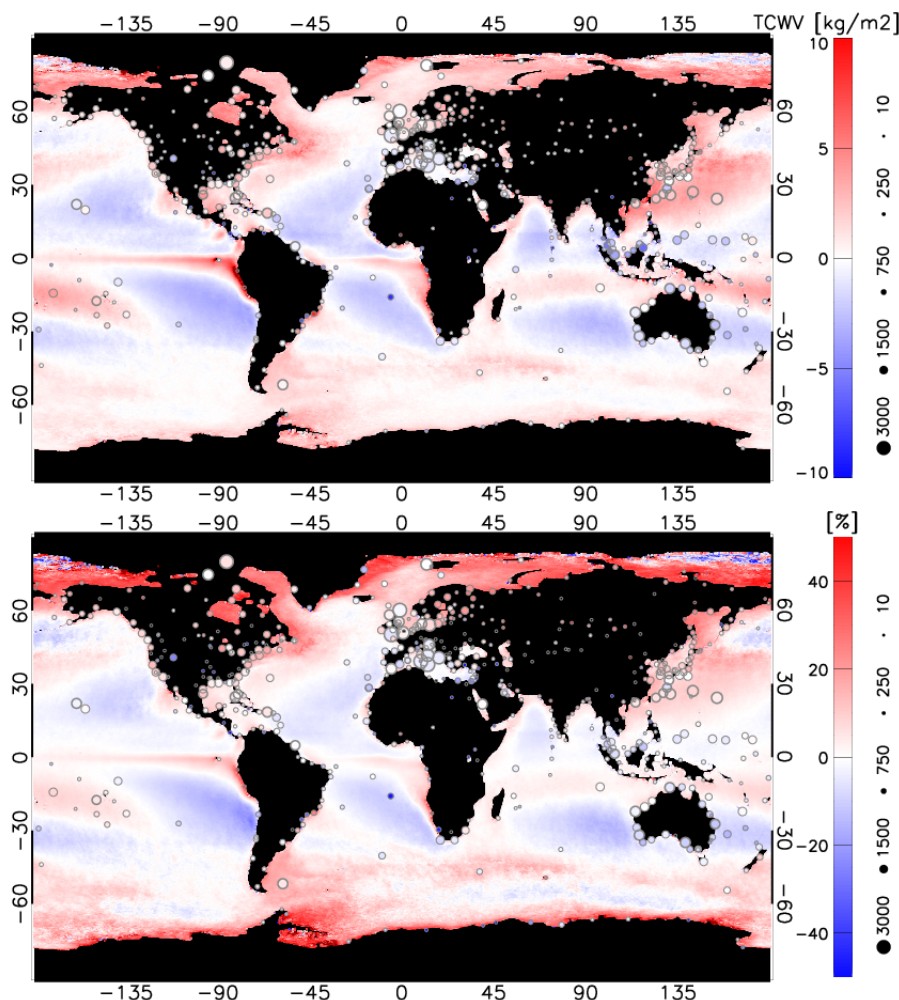

**Figure 7.** Average absolute (top) and relative (bottom) TCWV difference (SSM/I-AIRWAVE) at 0.25° × 0.25° spatial resolution. Average ARSA-AIRWAVE TCWV differences are overplotted with circles; the size of the symbols is proportional to the number of matches (see legend). The data cover the period from 1991 to 2012.

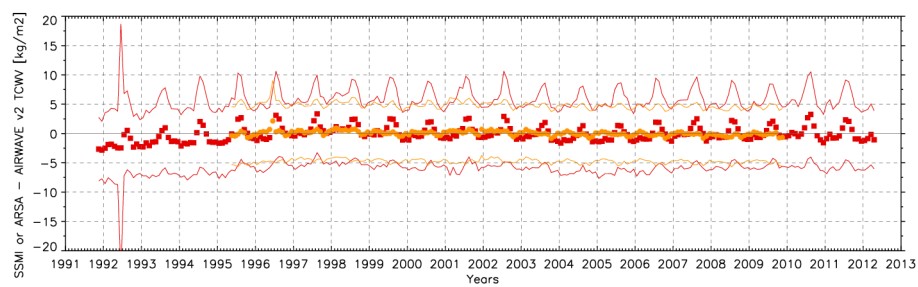

**Figure 8.** SSM/I-AIRWAVE TCWV (orange) and ARSA-AIRWAVE TCWV (red) monthly mean trends. The difference between the correlative measurements and AIRWAVE TCWV  STDV is also reported.

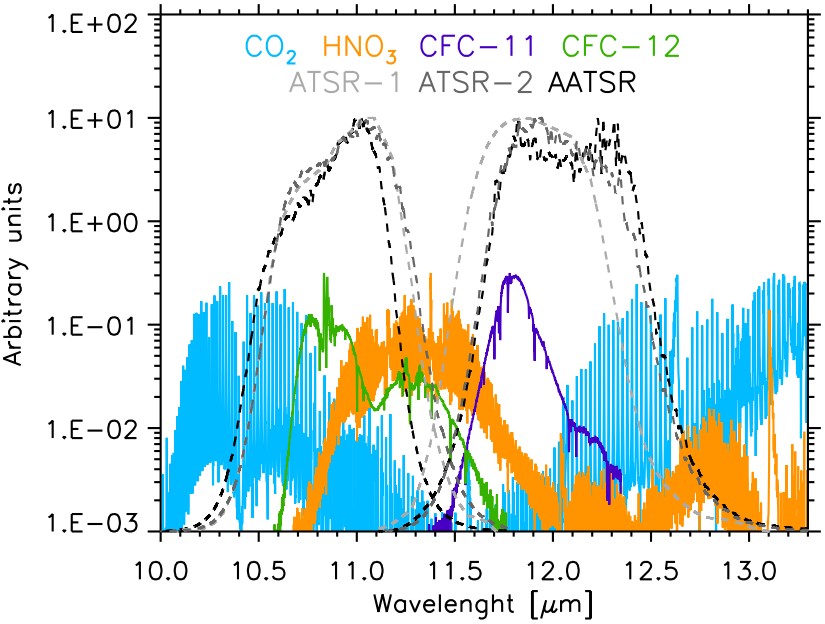

**Figure 9.** $CO_2$, $HNO_3$, CFCs spectral lines and ATSR filter functions (arbitrary units).