# Peer review of "The Advanced Infra-Red WAter Vapour Estimator (AIRWAVE) version 2: algorithm evolution, dataset description and performance improvements"

_Atmospheric Measurement Techniques, 2018_

## Referee Comment (RC1) · Anonymous Referee #1 · 27 Jul 2018

General Comments

This paper outlines updates made to the AIRWAVE algorithm that exploit the dual view of the ATSR series of IR radiometers to retrieve TCWV using the split window technique. The authors present seasonal maps of TCWV and present results from inter-comparisons with an established SSM/I TCWV product and ARSA radiosondes. The general comments were written after completing the specific and technical comments. When reviewing the results section the reader is redirected to a previous paper by the same group from 2018 for comparison of results. After downloading Papandrea et al. (2018) it becomes immediately apparent that the 2 manuscripts are very similar in lay-

out and appearance. The results from the earlier manuscript have practically identical figures, in the same order, with the only difference being Papandrea et al. (2018) validates the version 1 product whereas this study uses the new version 2 data. This study fails to show enough independence from the previously published work to warrant a new publication at this stage. Also, missing from the analysis/discussion is a quantification of the improvements between version 1 and 2 of the algorithm. Overall the study is of value due to the legacy of the ATSR series, and the FCDR the radiances represent. I would recommend this for publication only after all the issues that I have highlighted are addressed.

Specific Comments

1. Introduction lines 24-39: You mention microwave and near-infrared sensors, but what about water vapour from infrared sensors? TCWV estimates using the split window technique have been done with HIRS, AVHRR and MODIS to my knowledge. There is also no mention of challenges of ocean vs. land retrievals using IR window channels. What are the benefits of using the ATSR series?

2. Introduction line 42: Quantify 'general good quality' from previous assessment of version 1.

3. Section 2.1: Equation 1 it is unclear how lambda 1 & 2 are being used here. Is it referring to the 10.8 and 12 micron channels? How is it being used in the superscript notation? What is being multiplied with the optical depths? Also what is F? Further clarity is needed here.

4. Section 2.1 Line 96: Can you state the accuracy? Has this already been shown with AIRWAVE-v1?

5. Section 2.1 Line 136: Do the reported effects have an equal impact on retrieval precision for scan angles in the swath?

6. Section 2.2 line 156: So what year do you use and why?

7. Section 2.2 line 165: Why do you use ECMWF SSTs instead of the ARC/ESA CCI SST data products which are from the same instruments?

8. Section 3 line 194: Do you also retrieve TCWV over lakes?

9. Section 3 line 198: Are the uncertainties aggregated to the 0.25x0.25 grid? If so how are they propagated?

10. Figures 1-4: Too many sub-figures with replication of information. These should be combined into a single figure, removing either the 1b, 2b, 3b and 4b plots or the standard deviation maps to allow the reader to compare them side-by-side.

11. Section 3 line 223: Is this the RSS or HOAPS SSM/I product?

12. Section 3 line 234: Figure 7 is introduced before figure 6. Also you switch between Fig and Figure - please be consistent.

13. Figure 5: There is approximately 3 orders of magnitude more collocations with SSM/I than there are with ARSA. What impact does this have on the reported biases? Also the legends have different labels, the right-hand figure has bias while the left-hand figure says mean. Which is it? Ideally these should also be labelled a) and b).

14. Figure 6: If using colour filled regions that sit on top of one another then the alpha value needs to be lowered to add transparency so that both regions can be seen. Alternatively replace one with error bars.

15. Section 3 line 250: Do you require the reader to physically compare the table from Papandrea et al. 2018 with table 2 in this paper? From looking at the publication there is no table 2 but a table 1 and is this paper the validation of AIRWAVE-v1? This should be added to the discussion section if you want to make this comparison and discuss the improvements rather make the reader search them out.

16. Section 4: This section seems very empty especially as results from what assume to be the AIRWAVE-v1 product were only published earlier this year. There is a lack of

quantified improvements in the algorithm discussed or shown, especially as this is key to the title of the paper. Reads like a summary at best.

Technical Comments

1. Abstract line 1: First sentence doesn't read well. Suggested change to: "Total Column Water Vapour (TCWV) is a key atmospheric variable which is generally evaluated at global scales through the use of satellite data."

2. Introduction line 19: Remove the word 'the' after 'Actually,'

3. Introduction line 20: Full stop after Allen et al (2014) reference, begin new sentence with 'For this reason, ...'

4. Introduction line 44: Incorrect spelling: 'theese' (1 to many 'e')

5. The AIRWAVE v2 line 55: Delete the 'general' from 'general high quality'

6. The AIRWAVE v2 lines 55-63: You switch between AIRWAVE-v1, AIRWAVEv1 and v1

7. Section 2.1: Inconsistency in how AIRWAVE is referenced between v1 and v2 throughout section.

8. Section 2.1 Lines 95-96: 'is envisaged' suggests an aspirational future outcome. If this was done in AIRWAVE-v1 then it should be known whether this is true. The 2 sentences don not read well as the second sentence states that the known linear dependence allows for accurate retrievals. This is a little confusing to read, needs rewording.

9. Section 3 Line 192-193: Inconsistency in how AIRWAVE is referenced, here it is AIRWAVE V2 rather than AIRWAVE-v2 or V2. Need to settle on a single style.

10. Section 3 line 201: Same as above but now you use AIRWAVEv2

11. Section 3 line 224: DMSP already defined in Introduction

12. Section 3 line 239: incorrect spelling: 'radiosoundes' (no 'u')

13. Introduction & section 3: Acronym SSMI should be SSM/I.

---

## Referee Comment (RC2) · Anonymous Referee #2 · 31 Jul 2018

General comments

This paper describes a new version of the AIRWAVE algorithm and the retrieved Total Column Water Vapour (TCWV) product from the Along-Track Scanning Radiometer (ATSR) instruments. TCWV is obtained from the thermal infrared channels at 10.8 and 12 μm, the nadir and oblique views, and over sea surfaces. Validations are carried out based on aggregated products at 0.25 deg resolution and worldwide.

Overall, the reading of this paper leaves me in two minds. On the one hand, the value is principle very high due to the legacy of the very long time series of dataset provided by the ATSR series and the importance of deriving TCWV. The retrieval approach seems interesting too with the decomposition of the inverse model into a linear analytical equation. And the apparent evaluation results given at the end seem encouraging for this product and associated retrieval approach. But on the other hand, the added value and independency of this new manuscript w.r.t the requirements of the AMT journal and work already done and presented in the pre-existing studies, on which this manuscript relies, seem relatively little. I cannot at this stage recommend publication in AMT, and would like first to encourage the authors to address the major comments listed below:

- As emphasized by Referee #1, figures and layout of this manuscript are almost identical to the paper of Papandrea *et al.* (2018). The main difference is the evaluation of the v1 dataset, while here the focus is on v2. Apart of the generation of v2 (and associated new simulations), the present paper does not provide new works and innovations compared to the mentioned previous works.
- In several parts of this manuscript, the writing is a bit too qualitative, and even sometimes a bit ambiguous. Efforts shall be made to add more quantitative elements.
- The reader is sort out invited to check the improvements between AIRWAVE v1 and v2 by himself with the few lines given on Page 9. This is a bit unusual. A direct and explicit inter-comparison shall be provided here, with an explicit estimation of the bias reduction and precision improvements worldwide, per areas, and for different conditions.
- One of the expected improvements is the across-track variability. However, this element is not quantified in terms of uncertainty reduction on the TCWV. Same regarding the latitude dependency. Also please clarify sometimes if you are talking about bias, precision, overall uncertainty, etc…
- To warrant a new publication and invite the use of this new dataset, I think this manuscript should go beyond the previous ones by providing more validation exercises with new additional TCWV dataset instead of repeating similar work: from other satellite instruments: e.g. MODIS (Diedrich *et al.*, 2015), MERIS (*e.g.* Lindstrot *et al.*, 2012), ENVISAT Radiometer, MSG SEVIRI, and / or ground-based sensors (AERONET, GNSS, or other radiosondes), and / or ECMWF reanalysis. The approach proposed by AIRWAVE shall critically be analysed and compared with more "classically achieved" with other thermal infrared sensors to be in line with the ambitious title. Furthermore, what about the different sensitivities to the atmospheric layers of the different spectral ranges (*i.e.* thermal infrared and microwave). Does it allow, prevent, or limit in a certain range the comparison of the associated total columns?
- The evolutions in the AIRWAVE equations in Sect. 2.1., *i.e.* "latitude dependencies and across-track variations", do not explicitly appear. It is quite hard for the reader to be able to make the link between these, the given variables, and their physical meanings. The authors shall better

help the reader to establish these connections. Furthermore, I am a bit surprise that across-track and along-track viewing angles were not already considered in the previous version. And adding this in the update seems quite natural I think. As far as I can see, every atmospheric retrieval approach based on satellite sensors always needs to consider the sensor geometries to characterize accurately the average light path followed by the detected photons. Given how the air mass thickness varies with the geometry, the consideration of the viewing angles does not sound as very innovative to me, but rather quite natural. Other (precipitable) water retrieval from ATSR2, *e.g. Li et al.*, 2003, although over land, Ren *et al.* (2015) or even from SEVIRI (*e.g.* Sobrino *et al.*, 2007) already considered the variations of zenith angle at the surface for both nadir and forward views.

Specific comments

1. Abstract line 4: "performs the TCWV retrieval" => Please rephrase more direct *e.g.* "retrieves TCWV"
2. Abstract line 5: "combining nadir and forward observation geometries": The verb "combine, here and elsewhere in the manuscript, sounds overstated for me. Both geometries are not considered strictly combined in a way it helps to retrieve 1 single TCWV value, But both views are considered individually and independently for deriving TCWV per observation geometry…
3. Abstract line 7: "almost no bias" => Please be more quantitative and less qualitative. What does it mean "no bias"? I do not know any retrieval with an exactly null bias.
4. Abstract line 9: "these problems" => Which ones?
5. Abstract line 14: "significant improvements…" => Again, be quantitative. How much is the improvement overall? RMSE not defined previously. Clarify overall that AIRWAVE is over sea, no land!
6. Introduction line 36: Why AIRWAVE v1 is only available aggregated at the coarse resolution of 2x2 deg, and not for all the individual retrievals?
7. Introduction line 38 "good results" & line 42 "good quality": again, please be more specific and quantify. What the range of bias and precision for which you consider this is a good result?
8. Introduction line 44: "by accounting for latitudinal and angular variations of the retrieval": Not clear, retrieval estimations always vary with respect to geometry and latitude variation of the airmass. You probably mean the dependency of your forward & fitted model?
9. Introduction line 48: "spread" => Do you mean precision? Or uncertainty? Or spread of the differences with respect to another dataset?
10. Section 2 line 58 "average retrieval parameters": Which ones are you talking about here? Please be more specific. At this stage, the read haven't read it the equations in Sect. 2.1…
11. Section 2 line 66: again please clarify. Retrievals always vary with seasons and latitudes (water vapour properties). Do you mean that you explicitly consider the spatial and temporal variability of the some of the input parameters (e.g. $H_2O$ & temperature profiles)?
12. Section 2 line 70 "We recall': Was never said earlier in this paper.
13. Section 2.1 line 83 "F includes the atmospheric and surface radiance contribution". What is exactly F? This is not clear. Do you mean this is related to the temperature profile? Sea surface emissivity is already included later in the equation, so what is left? The temperature surface?
14. Section 2.1 line 93 "relative effective absorption cross-section": what do relative and effective mean here? Do you mean the absorption cross-section integrated along the average light path (by opposition to the vertical atmospheric layers)?
15. Section 2.1 line 104 "We verified…": Where is it shown? Please support your claimed verifications with adequate figures to convince the reader. Quantify this linearity (e.g. high correlation coefficient value).
16. Section 2.1 line 110 "average of all the G values obtained with different water vapour content": Please clarify the series of values for $H_2O$ content that you used or the list of

atmospheric conditions. Where do appear the zenith angles? They should already be, in theory, in G. J, emissivities, and F no?

17. Section 2.1 line 117 "sort of effective water vapour cross section": Please reformulate more properly. Such a description is rather vague and ambiguous. What do you mean by effective here?

18. Section 2.1 Eq.s 10 et al: Please be more specific by adding clearly all the angles in each equation: each view has a specific viewing angle. Furthermore, do you consider exactly the right zenith angle per pixel sensor? Or do you have a kind of average zenith angle for nadir and forward views? The Nadir view (*cf.* "NAD") cannot have only 1 zenith angle, or I miss something…

19. Section 2.1 line 136 "direct effect on the retrieval precision": And accuracy as well?

20. Section 2.2 line 173 "fixed wind speed": Why is it fixed? Why this value?

21. Section 2.2 line 235 and further: "a good correlation is obtained against both datasets, as highlighted by the correlations and bias values reported in the same figure". Please don't let the reader thinking and looking by himself for these numbers. Report them adequately in the relevant paragraphs.

22. Section 2.2 line 260: How the new algorithm reduces the impact of the sensor noise? This is not explained before. Any evidence?

23. Section 2.3 lines 184-185: I don't fully understand here. What do you interpolate exactly? I guess that now for each individual sensor pixel, you use the adequate zenith angle right?

24. Section 4 and appendix: Section 4 deserves more quantitative results with rigorous evaluation and validation of the AIRWAVE v2 to be in line with the title. More dataset, and more quantitative analyses per scan angle (since this is one of the claimed improvement). Also, how well does this approach achieve w.r.t to more classical approaches considered on thermal infrared sensors (e.g. Landsat, METEOSAT, etc…)? Moreover, I don't understand why Section 5 is here reported as an appendix. Some claims are written in the conclusion section, but were never reported before. I had to find out that some additional works were done in this appendix which is not obvious. Also what about the impact of the sea temperature surface? And H2o profile? Does it play a rule somewhere?

25. Table 1: What are the considered zenith angles considered here for the nadir and forward views?

26. Table 3: Please check and correct the units. Wind speed and temperature profile cannot be in kg/m2.

27. Apart of illustrations, what are the purposes of Figs1-4? I don't see any additional validations with them? Seem they take space with a lot of redundant information. I would advise to group them in 1 single plot with 4 panels. So then you can add more validations that would be kore relevant for this paper.

28. Fig5: What are the differences between "mean" (left panel) and "bias" (right panel)?

29. Figs5-6-7: Please precise the time period associated with all these figures. And the considered areas for Fig.5.

Proposed additional bibliography

Zhao-Liang Li, Li Jia, Zhongbo Su, Zhengming Wan & Renhua Zhang (2003) A new approach for retrieving precipitable water from ATSR2 split-window channel data over land area, International Journal of Remote Sensing, 24:24, 5095-5117, DOI: 10.1080/0143116031000096014

J. A. Sobrino & M. Romaguera (2008) Water-vapour retrieval from Meteosat 8/SEVIRI observations, International Journal of Remote Sensing, 29:3, 741-754, DOI: 10.1080/01431160701311267

Lindstrot, R., Preusker, R., Diedrich, H., Doppler, L., Bennartz, R., and Fischer, J.: 1D-Var retrieval of daytime total columnar water vapour from MERIS measurements, Atmos. Meas. Tech., 5, 631-646, https://doi.org/10.5194/amt-5-631-2012, 2012.

---

## Author Comment (AC1) · 10 Oct 2018

**General Comments**
This paper outlines updates made to the AIRWAVE algorithm that exploit the dual view of the ATSR series of IR radiometers to retrieve TCWV using the split window technique. The authors present seasonal maps of TCWV and present results from inter- comparisons with an established SSM/I TCWV product and ARSA radiosondes. The general comments were written after completing the specific and technical comments. When reviewing the results section the reader is redirected to a previous paper by the same group from 2018 for comparison of results. After downloading Papandrea et al. (2018) it becomes immediately apparent that the 2 manuscripts are very similar in layout and appearance. The results from the earlier manuscript have practically identical figures, in the same order, with the only difference being Papandrea et al. (2018) validates the version 1 product whereas this study uses the new version 2 data. This study fails to show enough independence from the previously published work to warrant a new publication at this stage.

The main concern of the reviewer is related to the fact that the present paper seems not enough independent from Papandrea et al., 2018, that contains AIRWAVEv1 validation. He/she says that this is due to the fact that the lay-out of figures is the same in the two papers.
The reviewer is correct about the style of the figures regarding the validation. However, this is not true for the content of these figures! The use of the same figures layout is meant to produce a benefit for the reader, who can easily appreciate the differences between the two versions of the algorithm. Furthermore, the present paper contains the description of AIRWAVEv2 algorithm and a climatology of the dataset in addition to validation. We think that all these elements guarantee enough independency from previous work and that the improvements of version 2, are relevant and deserve publication.
In order to clarify the relevance of these improvements, also considering the other reviewer's comments, we add information on the comparisons between AIRWAVEv1 and AIRWAVEv2 (see specific comments below and reply to reviewer #2).

Also, missing from the analysis/discussion is a quantification of the improvements between version 1 and 2 of the algorithm.

In the revised version of the paper we report directly the comparison of performances of AIRWAVE version 1 and version2 adding in Table 2 the results of AIRWAVEv1 validation from Papandrea et al., 2018. This allow the direct comparison and the quantification of the improvements between the two version of the algorithm. The results of the comparisons and the improvements obtained with AIRWAVEv2 are also reported in session "Discussion and Conclusions".

Overall the study is of value due to the legacy of the ATSR series, and the FCDR the radiances represent. I would recommend this for publication only after all the issues that I have highlighted are addressed.

**Specific Comments**
1. Introduction lines 24-39: You mention microwave and near-infrared sensors, but what about water vapour from infrared sensors? TCWV estimates using the split window technique have been done with HIRS, AVHRR and MODIS to my knowledge. There is also no mention of challenges of ocean vs. land retrievals using IR window channels. What are the benefits of using the ATSR series?

To cover this point, in the revised version of the paper we add:

"TCWV retrievals from infrared spectral regions were performed from Advanced Very High Resolution Radiometer (AVHRR; Emery, 1992) measurements, using the split window technique, (Sobrino et al., 1991) and from MODIS (Seeman et al., 2003). TCWV retrievals from infrared channels over land suffer of the limited knowledge of the temperature and the emissivity of land surfaces (Lindstrot et al., 2014)"

before "The Along-Track Scanning Radiometer (ATSR, Delderfield et al. (1986)) instrument series had as main objective the accurate ..."

In line 37 we also add, following the reviewer's suggestion "Due to the legacy of the ATSR series, and the fact that the radiances are a fundamental climate dataset record, the AIRWAVE dataset is an important resource for water vapor studies."

Consequently, we add the following references:

Lindstrot, R. M. Stengel, M. Schröder, J. Fischer, R. Preusker, N. Schneider, T. Steenbergen, and B. R. Bojkov. "A global climatology of total columnar water vapour from SSM/I and MERIS", Earth Syst. Sci. Data, 6, 221–233, 2014 www.earth-syst-sci-data.net/6/221/2014/ doi:10.5194/essd-6-221-2014

Seemann, S., J. Li, W. P. Menzel, and L. Gumley (2003), Operational retrieval of atmospheric temperature, moisture, and ozone from MODIS infrared radiances, J. Appl. Meteorol., 42, 1072–1091.

Sobrino, J. A.; Coll, C.; Caselles, V. Atmospheric corrections for land surface temperature using AVHRR channel 4 and 5. Remote Sens. Environ. 1991, 38, 19-34.

2. Introduction line 42: Quantify 'general good quality' from previous assessment of version 1.

Done. we add "(average correlative bias of 0.72 kg/m2 vs SSM/I and 0.80 kg/m2 vs ARSA)" after "general good quality of AIRWAVEv1 dataset".

3. Section 2.1: Equation 1 it is unclear how lambda 1 & 2 are being used here. Is it referring to the 10.8 and 12 micron channels? How is it being used in the superscript notation? What is being multiplied with the optical depths? Also what is F? Further clarity is needed here.

Yes, lambda 1 refers to the channel at 10.8 um, while lambda 2 is for the channel at 12 um. In the superscript it is the exponent of the term. The optical depths are multiplied by the values of the frequencies. F is now explicitly described and in the revised version of the paper we clarify all these points.

4. Section 2.1 Line 96: Can you state the accuracy? Has this already been shown with AIRWAVE-v1?

In Casadio et al., 2016 Appendix A we evaluate the accuracy of geometric correction to the AMF for the TIR channels of ATSR. In this paper we evaluate the accuracy of the whole retrieval procedure (not only the linear dependence between TCWV and optical depths).

5. Section 2.1 Line 136: Do the reported effects have an equal impact on retrieval precision for scan angles in the swath?

The estimates reported in the paper are given for along track configuration only. In the revised version of the paper we specify this in the text. However, we can also give an estimate of the improvement for extreme across track points: In worse cases, the precision has at maximum 1% higher value in the extreme across track of the swath with respect to the sub satellite points. We add this information at the end of Section 2.1.

6. Section 2.2 line 156: So what year do you use and why?

We use the 2010 (we added this information in the revised version of the paper). As stated in the text, the year to year variability has globally no impact on the spectra apart from some species whose impact has been assessed in the appendix.

7. Section 2.2 line 165: Why do you use ECMWF SSTs instead of the ARC/ESA CCI SST data products which are from the same instruments?

The SST used to calculate the retrieval parameter should be just representative of average conditions in a given season and latitude band. For this scope average SSTs obtained from the easily accessible ECMWF monthly means are suitable. ARC/ESA SSTs would have been suitable too.

8. Section 3 line 194: Do you also retrieve TCWV over lakes?

Yes, we clarify this in the revised version of the paper. We add "surfaces (sea and lakes)" after "over water"

9. Section 3 line 198: Are the uncertainties aggregated to the 0.25x0.25 grid? If so how are they propagated?

No, As specified in lines 199-200 the SSMI group contains also the values of standard deviations for each 0.25x0.25 grid points. In the revised version of the paper we rephrase in order to clarify this point.
We replace :"The SSM/I group, in addition, contains the value of the standard deviation and the number of elements aggregated within the SSM/I grid cell. "
with
"The SSM/I group, in addition, contains the value of the number of elements aggregated within the SSM/I grid cell and the standard deviation of the TCWV value associated to each cell. "

10. Figures 1-4: Too many sub-figures with replication of information. These should be combined into a single figure, removing either the 1b, 2b, 3b and 4b plots or the standard deviation maps to allow the reader to compare them side-by-side.

Following also the comments by Reviewer #2 we combine in one single figure figures 1,2,3 and 4 a and 1,2,3,4 c, now it is Figure 2.
Figures b) were moved in a separate figure, now it is Figure 3. We changed the text and the captions accordingly.

11. Section 3 line 223: Is this the RSS or HOAPS SSM/I product?

RSS as in Papandrea et al. 2018. This information has been added into the revised version of the paper where we describe with version of the SSM/I dataset was used for the comparison.

12. Section 3 line 234: Figure 7 is introduced before figure 6. Also you switch between Fig and Figure - please be consistent.

In the revised text we have switched the order of figures 5 and 6. We use Figure when it is at the beginning of a sentence and Fig. in all other cases as specified in AMT guidelines.

13. Figure 5: There is approximately 3 orders of magnitude more collocations with SSM/I than there are with ARSA. What impact does this have on the reported biases? Also the legends have different labels, the right-hand figure has bias while the left-hand figure says mean. Which is it? Ideally these should also be labelled a) and b).

About the different number of collocations: The reviewer is right when he says that the satellite has 3 orders of magnitude collocations than the radiosondes. However the number of collocations is very high (order of 10^5) and thus the impact on the bias is not related to that. However, the positions of the ARSA stations, that are not equally distributed worldwide as the satellite measurements (the majority of sites is based at mid-latitudes), may result in a small displacement of the TCWV values that are compared w.r.t. satellite (i.e. less relative amount of very high TCWV, characteristic of tropical conditions).
Done. We add a) and b) in figures and use "bias" in both panels (now it is Figure 5).

14. Figure 6: If using colour filled regions that sit on top of one another then the alpha value needs to be lowered to add transparency so that both regions can be seen. Alternatively replace one with error bars.

Done. In the revised version AIRWAVEv1 has blue error bars (now it is Figure 4).

15. Section 3 line 250: Do you require the reader to physically compare the table from Papandrea et al. 2018 with table 2 in this paper? From looking at the publication there is no table 2 but a table 1 and is this paper the validation of AIRWAVE-v1? This should be added to the discussion section if you want to make this comparison and discuss the improvements rather make the reader search them out.

Done.We added the values of bias and standard deviations of the differences between AIRWAVEv1 and SSM/I , ARSA extracted from Table 1 of Papandrea et al., 2018 in Table 2. We also change Table caption and text accordingly.

16. Section 4: This section seems very empty especially as results from what assume to be the AIRWAVE-v1 product were only published earlier this year. There is a lack of quantified improvements in the algorithm discussed or shown, especially as this is key to the title of the paper. Reads like a summary at best.

In the revised version of the paper we add here a quantification of the improvements obtained from version 1 to version 2.
We added:
As expected also from the analysis of synthetic retrievals, the most significant AIRWAVEv2 improvement is achieved at polar latitudes. In polar regions the bias versus SSM/I improves of 4.2 kg/m2 and of 3.2 kg/m2 versus ARSA. In both cases the standard deviations are reduced of about 1.6-1.9 kg/m2 .However, improvements at mid-latitudes are also found. The average bias with respect to SSM/I improves of about 0.7 kg/m2 and the standard deviation is reduced of about 1 kg/m2. In case of validation against radiosondes the bias in AIRWAVEv2 is reduced of about 0.6 kg/m2 with respect to AIRWAVEv1 and the standard deviation is reduced of 1.6 kg/m2."

**Technical Comments**
1. Abstract line 1: First sentence doesn't read well. Suggested change to: "Total Column Water Vapour (TCWV) is a key atmospheric variable which is generally evaluated at global scales through the use of satellite data."

Done.

2. Introduction line 19: Remove the word 'the' after 'Actually,'

Done.

3. Introduction line 20: Full stop after Allen et al (2014) reference, begin new sentence with 'For this reason, …'

Done.

4. Introduction line 44: Incorrect spelling: 'theese' (1 to many 'e')

Done.

5. The AIRWAVE v2 line 55: Delete the 'general' from 'general high quality'

Done.

6. The AIRWAVE v2 lines 55-63: You switch between AIRWAVE-v1, AIRWAVEv1 and v1

In the revised version of the paper we use AIRWAVEv1 everywhere.

7. Section 2.1: Inconsistency in how AIRWAVE is referenced between v1 and v2 throughout section.

In the revised version of the paper we use AIRWAVEv1 everywhere.

8. Section 2.1 Lines 95-96: 'is envisaged' suggests an aspirational future outcome. If this was done in AIRWAVE-v1 then it should be known whether this is true. The 2 sentences don not read well as the second sentence states that the known linear dependence allows for accurate retrievals. This is a little confusing to read, needs rewording.

Done. In the revised version of the paper "is envisaged" is removed and the two sentences were rephrased: "This equation shows that a linear behavior exists between the water vapour optical depth and the TCWV.
The linear dependence is exploited to solve the AIRWAVE equation and to retrieve TCWV. "

9. Section 3 Line 192-193: Inconsistency in how AIRWAVE is referenced, here it is AIRWAVE V2 rather than AIRWAVE-v2 or V2. Need to settle on a single style.

In the revised version of the paper we use AIRWAVEv2 everywhere.

10. Section 3 line 201: Same as above but now you use AIRWAVEv2 11. Section 3 line 224: DMSP already defined in Introduction

In the revised version of the paper we use AIRWAVEv2 everywhere. DMSP Done.

12. Section 3 line 239: incorrect spelling: 'radiosoundes' (no 'u')

Done.

13. Introduction & section 3: Acronym SSMI should be SSM/I.

Done.

---

## Author Comment (AC2) · 10 Oct 2018

**General comments**

This paper describes a new version of the AIRWAVE algorithm and the retrieved Total Column Water Vapour (TCWV) product from the Along-Track Scanning Radiometer (ATSR) instruments. TCWV is obtained from the thermal infrared channels at 10.8 and 12 µm, the nadir and oblique views, and over sea surfaces. Validations are carried out based on aggregated products at 0.25 deg resolution and worldwide.
Overall, the reading of this paper leaves me in two minds. On the one hand, the value is principle very high due to the legacy of the very long time series of dataset provided by the ATSR series and the importance of deriving TCWV. The retrieval approach seems interesting too with the decomposition of the inverse model into a linear analytical equation. And the apparent evaluation results given at the end seem encouraging for this product and associated retrieval approach. But on the other hand, the added value and independency of this new manuscript w.r.t the requirements of the AMT journal and work already done and presented in the pre-existing studies, on which this manuscript relies, seem relatively little. I cannot at this stage recommend publication in AMT, and would like first to encourage the authors to address the major comments listed below:

• As emphasized by Referee #1, figures and layout of this manuscript are almost identical to the paper of Papandrea et al. (2018). The main difference is the evaluation of the v1 dataset, while here the focus is on v2. Apart of the generation of v2 (and associated new simulations), the present paper does not provide new works and innovations compared to the mentioned previous works.

As replied to reviewer #1, it is true that the figures have the same outline of Papandrea et al 2018. This was done to ease the comparison between the two dataset versions. However, the content of the figures is totally different! and also the layout of the paper is not the same. In this paper we present the new version of the algorithm with all the new calculations used to improve the retrieval performances. Then we show the performances of the new dataset against the AIRWAVEv1 and we present the climatology from AIRWAVEv2 dataset. In addition, we provide a quantification of major error sources. Due to the entity of the improvements of the TCWV AIRWAVEv2 dataset we think that the paper presents new work with respect to the previous ones.

• In several parts of this manuscript, the writing is a bit too qualitative, and even sometimes a bit ambiguous. Efforts shall be made to add more quantitative elements.

Done. Following the reviewer's suggestions, we add more quantification of the results and clarified the points raised by the reviewer.

• The reader is sort out invited to check the improvements between AIRWAVE v1 and v2 by himself with the few lines given on Page 9. This is a bit unusual. A direct and explicit inter- comparison shall be provided here, with an explicit estimation of the bias reduction and precision improvements worldwide, per areas, and for different conditions.

Done. We report the results of the validation of AIRWAVEv1 in Table 2 together with AIRWAVEv2. Then we add a discussion on that and a direct comparison in Section 4.

- One of the expected improvements is the across-track variability. However, this element is not quantified in terms of uncertainty reduction on the TCWV. Same regarding the latitude dependency. Also please clarify sometimes if you are talking about bias, precision, overall uncertainty, etc...

In the revised version of the paper we clarify when we talk about bias or precision. Regarding the across track variability a more extensive reply is given below. In Appendix A we give an estimate of the accuracy of the new parameters using simulated values. When applied to real data, due to the nature of the algorithm and to the used interpolations, is not easy to decouple the single effects on the retrieval. However, in the paper we report the results (in terms of bias and standard deviation) of the validation against other instruments also for latitude regions in order to highlight the impact of the new algorithm in different latitude bands.

- To warrant a new publication and invite the use of this new dataset, I think this manuscript should go beyond the previous ones by providing more validation exercises with new additional TCWV dataset instead of repeating similar work: from other satellite instruments: e.g. MODIS (Diedrich et al., 2015), MERIS (e.g. Lindstrot et al., 2012), ENVISAT Radiometer, MSG SEVIRI, and / or ground-based sensors (AERONET, GNSS, or other radiosondes), and / or ECMWF reanalysis. The approach proposed by AIRWAVE shall critically be analysed and compared with more "classically achieved" with other thermal infrared sensors to be in line with the ambitious title. Furthermore, what about the different sensitivities to the atmospheric layers of the different spectral ranges (i.e. thermal infrared and microwave). Does it allow, prevent, or limit in a certain range the comparison of the associated total columns?

The main scopes of this paper is to describe the algorithm evolution, the dataset and performance improvements. For these reasons we repeat the validation exercise using the same validation dataset used for AIRWAVEv1. Furthermore, the validation is performed using satellite microwave data and radiosondes data, thus two very different datasets (platforms, sensitivity …) allowing to evaluate the AIRWAVEv2 performances under two very different point of view. In future, additional validation exercises will be performed (e.g. with MWR on ENVISAT), however this is out of the scope of the present paper.
Regarding the different sensitivities of TIR and microwave, in the validation exercise we compare SSM/I data in coincidence with AIRWAVEv2 ones. Since AIRWAVE is applicable only to clear sky measurements this is a method to filter out SSM/I "cloudy" TCWV and thus to avoid biases due to different sensitivity related to the used spectral range. When comparing to radiosondes, the small bias we found demonstrate that AIRWAVE TCWV are sensitive also to low atmospheric levels.

In the revised version of the paper we added a sentence about the different sensitivity of TIR, microwave and radiosondes reported above.

- The evolutions in the AIRWAVE equations in Sect. 2.1., i.e. "latitude dependencies and across-track variations", do not explicitly appear. It is quite hard for the reader to be able to make the link between these, the given variables, and their physical meanings. The authors shall better help the reader to establish these connections. Furthermore, I am a bit surprise that across- track and along-track viewing angles were not already considered in the previous version. And adding this in the update seems quite natural I think. As far as I can see, every atmospheric retrieval approach based on satellite sensors always needs to consider the sensor geometries to characterize accurately the average light path followed by the detected photons. Given how the air mass thickness varies with the geometry, the consideration of the viewing angles does not sound as very innovative to me, but rather quite natural. Other (precipitable) water retrieval from ATSR2, e.g. Li et al., 2003, although over land, Ren et al. (2015) or even from SEVIRI (e.g. Sobrino et al., 2007) already considered the variations of zenith angle at the surface for both nadir and forward views.

In the revised version of the paper we follow the reviewer's suggestion reported also in the specific comments and tried to clarify the quantities in the equations, their link with viewing geometries and physical quantities.

Regarding the across track angles: possibly there is a misunderstanding here. In Casado et al., 2016 : "To account for the variability of the nadir view angle (0–25°), the TCWV evaluated through Eq. (29) are corrected using an empirical correction factor, dependent on the across track index position of the ATSR measurements. This across-track correction has been evaluated through radiative transfer simulation and provides a sufficiently accurate but extremely fast solution (in terms of CPU efficiency) to this problem. In the next version of the AIRWAVE algorithm, the retrieval parameters will be extracted from pre-calculated look-up tables, in which the angular dependence will be considered." And what we present in this paper is exactly the calculation of angular dependent retrieval parameters.

[Figure]

**Figure 0: Differences between TCWV calculated from across pixels and TCWV calculated sub-satellite track as a function of across track position calculated for different atmospheric scenarios and for AATSR. Black dotted line corresponds to AIRWAVEv1 parametrization.**

Figure 0 shows the absolute difference between TCWV from across track pixels and sub-satellite pixels calculated for AATSR in different atmospheric scenarios (tropical and mid-latitude) from synthetic measurements. To produce these retrievals, we simulated ATSR(s) radiances at the eleven tie points for the nadir and forward views considering also the surface emissivity variations with viewing angles. The atmospheric scenario, the TCWV and the SST were kept constant and they were exactly the same used for the sub-satellite track case. Then we retrieved the TCWV for each of the 11 couples of BTs and computed the difference with respect to the value obtained at the sub-satellite track position (that coincide with the TCWV reference value). The dotted line in Fig. 0 mimics the correction term adopted within the AIRWAVEv1 algorithm. As can be seen the dotted line well mimic the general behavior of the across track TCWV dependence. This is confirmed by the comparison made between AIRWAVEv1 TCWV and SSMI or ECMWF TCWV (see Casadio et al., 2016): on average we did not found any across-track bias, thus confirming the general validity of this correction. However, as can be seen in Fig. 0, in specific cases this approximation can generate differences depending on the used atmospheric scenario. Furthermore, a slight asymmetry with respect to the along track position is also visible. This behavior is expected as the ATSR instruments are tilted of about 4 respect to the flying direction of the satellite. In order to properly account for these variations, in AIRWAVEv2 we replace the a-posteriori correction by directly calculating the retrieval parameters for each of the 11 tie points of the nadir and forward swaths and interpolating the results at each position of the ground pixels. We also use in the solving equations the

correct emissivity values, in order to properly take into account the angular dependence of the sea surface emissivity.

In the revised version of the paper we add some details to clarify this (see replies to specific comments # 8,18 and 23).

**Specific comments**
1.  Abstract line 4: "performs the TCWV retrieval" => Please rephrase more direct e.g. "retrieves TCWV"

Done.

2. Abstract line 5: "combining nadir and forward observation geometries": The verb "combine, here and elsewhere in the manuscript, sounds overstated for me. Both geometries are not considered strictly combined in a way it helps to retrieve 1 single TCWV value, But both views are considered individually and independently for deriving TCWV per observation geometry…

We remove "combining" in the abstract. Then we use "using" in other part (line 31 of the discussion version of the paper).

3. Abstract line 7: "almost no bias" => Please be more quantitative and less qualitative. What does it mean "no bias"? I do not know any retrieval with an exactly null bias.

We rephrase "very good agreement with almost no bias all over the ATSR missions, with the exception of the polar and the coastal region where AIRWAVE underestimate the TCWV amount." —> "very good agreement with an overall bias of 3% all over the ATSR missions. A large contribution to this bias comes from the polar and the coastal region where AIRWAVE underestimate the TCWV amount."

4. Abstract line 9: "these problems" => Which ones?

The underestimation of TCWV in coastal and polar areas. We change "to overcome these problems" with "to reduce the bias in these regions".

5. Abstract line 14: "significant improvements..." => Again, be quantitative. How much is the improvement overall? RMSE not defined previously. Clarify overall that AIRWAVE is over sea, no land!

Done. Now it is: "Results show significant improvements in both biases (from 0.72 to 0.02 kg/m2) and standard deviations (from 5.75 to 4.69 kg/m2 versus SSM/I)". We add "over sea" in the fourth abstract sentence: "The algorithm was used to produce a TCWV database over sea from the whole ATSR mission".

6. Introduction line 36: Why AIRWAVE v1 is only available aggregated at the coarse resolution of 2x2 deg, and not for all the individual retrievals?

AIRWAVE is included into the G-VAP at this resolution. Contacting the authors, the complete dataset can be obtained. Actually on the G-VAP only monthly means from 2003-2008 are available.

7. Introduction line 38 "good results" & line 42 "good quality": again, please be more specific and quantify. What the range of bias and precision for which you consider this is a good result?

Done. We added: "(average correlative bias of 0.72 kg/m2 vs SSM/I and 0.80 kg/m2 vs ARSA, below the 1 kg/m2 indicated in the GlobVapour project (Lindstrot et al., 2010))" after This exercise demonstrated a general good quality of AIRWAVEv1 dataset
In the PVR of the GlobVapour project report they use as goal 1kg/m2 for bias. Results from both AIWAVEv1 and AIRWAVEv2 are below this threshold.

8. Introduction line 44: "by accounting for latitudinal and angular variations of the retrieval": Not clear, retrieval estimations always vary with respect to geometry and latitude variation of the air-mass. You probably mean the dependency of your forward & fitted model?

First of all we want to clarify that AIRWAVE does not make direct use of a forward model, but uses pre-computed parameters (retrieval parameters) to solve equation 13 of the revised version of the paper. In AIRWAVEv1 we used fixed parameters along the globe, in addition, "to account for the variability of the nadir view angle (0–25°), the TCWV evaluated through Eq. (29) are corrected using an empirical correction factor, dependent on the across track index position of the ATSR measurements."(Casadio et al., 2016). In AIRWAVEv2 we use latitudinal dependent retrieval parameters calculated for each of the 22 (11 nadir and 11 forward) tie angles. To clarify this,in the revised version of the paper we add :
"AIRWAVEv1 use fixed retrieval parameters along the globe and TCWV are corrected for viewing angles variability in nadir and slant by using an empirical correction factor." in line 39 after " For this reason, in the V1 algorithm several approximations were made."

9. Introduction line 48: "spread" => Do you mean precision? Or uncertainty? Or spread of the differences with respect to another dataset?

Spread of the differences with respect to another dataset. We modified the text accordingly.

10. Section 2 line 58 "average retrieval parameters": Which ones are you talking about here? Please be more specific. At this stage, the read haven't read it the equations in Sect. 2.1…

We reformulate:" AIRWAVEv1 makes use of retrieval parameters calculated though RTM simulations of tropical and mid-latitude atmospheric scenarios then averaged and used for the whole globe."

11. Section 2 line 66: again please clarify. Retrievals always vary with seasons and latitudes (water vapour properties). Do you mean that you explicitly consider the spatial and temporal variability of the some of the input parameters (e.g. H2O & temperature profiles)?

We mean that we did not use average retrieval parameters as done in AIRWAVEv1, instead we compute the retrieval parameters for 6 latitude bands and 4 months. Then during the retrieval these are used as look-up-tables. In the revised version of the paper we rephrased:
"Secondly, we computed the retrieval parameters taking into account both their seasonal and latitudinal dependence" —> "Secondly, we compute the retrieval parameters for different latitude bands and for four months that, in the retrieval, are used as look-up-tables"

12. Section 2 line 70 "We recall': Was never said earlier in this paper.

We removed " We recall here that".

13. Section 2.1 line 83 "F includes the atmospheric and surface radiance contribution". What is exactly F? This is not clear. Do you mean this is related to the temperature profile? Sea surface emissivity is already included later in the equation, so what is left? The temperature surface?

In the revised version of the paper we clarified this by explicitly indicating what is F. See also reply to reviewer #1.

14. Section 2.1 line 93 "relative effective absorption cross-section": what do relative and effective mean here? Do you mean the absorption cross-section integrated along the average light path (by opposition to the vertical atmospheric layers)?

With effective here we mean that the cross section is multiplied for the lambda value. This is the same definition used in Casadio et al., 2016. In the revised version of the paper we replaced "effective absorption cross section (sigma)" with "effective absorption cross section (lambda*sigma)"

15. Section 2.1 line 104 "We verified...": Where is it shown? Please support your claimed verifications with adequate figures to convince the reader. Quantify this linearity (e.g. high correlation coefficient value).

To clarify this we add a figure in the revised version of the paper and we modify the text as follows:

[Figure]

**Figure 1: logarithm of radiance ratio at nadir as a function of TCWV in simulated atmospheric scenario. Grey dots represent the same quantity using real AATSR radiances and coincident SSMI TCWV for the along track measurements on the 5 and 6 August 2008.**

In Fig. 1 the colored dots represent the values of the logarithm of radiance ratio in equation (1) as a function of the TCWV for the different atmospheric scenarios. We report only the values obtained for the sub satellite scans using the IG2 water profiles for the Summer season multiplied for 0.5 and 1.5. The different colors represent different latitude bands (going from red for tropical to blue for polar). The grey dots represent the radiance ratio calculated from along track AATSR measurements on the 5 and 6 of August 2008 aggregated at SSMI resolution (0.25° 0.25°). The value of TCWV associated to each AATSR sub-satellite measurement was obtained from coincident SSMI measurements. In order to minimise the impact of random error, only measurements with SSMI pixel coverage (calculated as the ratio between the actual and the maximum number of ATSR measurements that can be present into a SSMI pixel) greater than 10% were used for this exercise. Figure 1 shows that: a) the simulated radiances correctly reproduce the real measurement behavior; b) the relation between the radiances and the TCWV can be considered as linear.

Actually, we find in this case a correlation of 0.904 for real data and 0.92 for the simulated ones (p-value of 7.2630294e-05).

16. Section 2.1 line 110 "average of all the G values obtained with different water vapour content": Please clarify the series of values for H2O content that you used or the list of atmospheric conditions. Where do appear the zenith angles? They should already be, in theory, in G. J, emissivities, and F no?

We added: "(to vary the water vapour content we multiplied the water vapor profile for 0.5, 0.75, 1., 1.25, 1.5)"
after
" For this reason, for each atmospheric scenario the average of all the G values obtained with different water vapour content is used".
For the zenith angles the reviewer is right, we added additional information on this, see reply to comments 8, 18 and 23.

17. Section 2.1 line 117 "sort of effective water vapour cross section": Please reformulate more properly. Such a description is rather vague and ambiguous. What do you mean by effective here?

We remove the sentence.

18. Section 2.1 Eq.s 10 et al: Please be more specific by adding clearly all the angles in each equation: each view has a specific viewing angle. Furthermore, do you consider exactly the right zenith angle per pixel sensor? Or do you have a kind of average zenith angle for nadir and forward views? The Nadir view (cf. "NAD") cannot have only 1 zenith angle, or I miss something...

From equation 12 onward we explicit the angular dependence. We calculate all the parameters at the 11 tie point positions. It means that we calculate the equations for 11 nadir angles (from 0 to 21 degrees) and 11 Forward angles (from 53 to 55.3 °). Then we interpolate using each single pixel across track position.
To clarify this, in the paper we added: "Equations (11), (12), (13) were solved for the 11 couples of viewing angles corresponding to the tie points. The angles cover a range from 0 to 21 degrees in the NAD case and from 53 to 55 degrees in the FWD case." after "Equations (10), (12) and (13) are the solving equations used in AIRWAVEv2, while AIRWAVEv1 makes use of equations (11), (12) and (13). "

19. Section 2.1 line 136 "direct effect on the retrieval precision": And accuracy as well?

A preliminary assessment of the accuracy of the new method can be found in Appendix A, Section 5.1 using simulated radiances.

20. Section 2.2 line 173 "fixed wind speed": Why is it fixed? Why this value?

Due to wind speed variability, in both algorithm version, we prefer to fix the value of the wind speed and then treat this as an error source (see section 5.3). However, as can be noticed, this impact is negligible in equatorial regions, very low (0.7%) in midlatitude regions and is about 2.7 % in polar regions when the wind speed reaches the high value of 25 m/s. We choose this value (3 m/s) as a reference because is a reasonable trade-off between 1 m/s and 25 m/s that are extreme values available in the University of Edinburgh emissivity database. The 3 m/s is quite close to the average wind speed over the sea that is about 6 m/s (e.g. Monhan: "The Probability Distribution of Sea Surface Wind Speeds. Part I: Theory and SeaWinds Observations", Journal of Climate, 19, 2006. or data from Copernicus Marine environment monitoring centre data on 100km monthly L$ 2007-2012 Climatology Global Wind).

In the revised paper in section 2.2 we add: "Due to wind speed variability, in both algorithm version, we prefer to fix the value of the wind speed and then treat wind variations as an error source (see Sect 5.3)", before the end of the section.

21. Section 2.2 line 235 and further: "a good correlation is obtained against both datasets, as highlighted by the correlations and bias values reported in the same figure". Please don't let the reader thinking and looking by himself for these numbers. Report them adequately in the relevant paragraphs.

Following the reviewer's suggestion in this section we changed:
"Globally, a good correlation is obtained against both datasets, as highlighted by the correlations and bias values reported in the same figure."
with
"Globally, a good correlation is obtained against both datasets, as highlighted by the correlations (0.948 with SSM/I and 0.918 with ARSA) and bias values (0.02 +/- 4.79 kg/m2 with respect to SSM/I and 0.19 +/- 6.12 kg/m2 with respect to ARSA)."

and
"The improvement in the performances of the new dataset is clearly visible at all latitudes."
with
"The improvement in the performances of the new dataset is clearly visible at all latitudes, and in particular for regions with latitude higher than 45-50° where the negative values obtained with AIRWAVEv1 disappear. In addition, a significant reduction of the spread is highlighted."

Then we add in Table 2 the results of the validation of AIRWAVEv1 as extracted from Papandrea et al., 2018.

22. Section 2.2 line 260: How the new algorithm reduces the impact of the sensor noise? This is not explained before. Any evidence?

It is at the end of Section 2.1 we evaluate the impact of the sensor noise using the new equations.

23. Section 2.3 lines 184-185: I don't fully understand here. What do you interpolate exactly? I guess that now for each individual sensor pixel, you use the adequate zenith angle right?

As stated before, we calculate the retrieval parameters for the 11 tie points. Then using the across track position of each sensor pixel we interpolate the retrieval parameters calculated for the 11 tie points angles at the exact pixel position (and zenith angles).
To clarify we replaced:" In V2 we replaced the a-posteriori correction by directly calculating the retrieval parameters for each of the above described tie points of the nadir and forward swaths and interpolating the results at each position of the ground pixels."
with:
"In AIRWAVEv2 we replaced the a-posteriori correction: We calculated the retrieval parameters for each of the above described tie points of the nadir and forward swaths, then we obtained the parameters at the exact ground pixel position interpolating these values and using the ground pixel across track position."

24. Section 4 and appendix: Section 4 deserves more quantitative results with rigorous evaluation and validation of the AIRWAVE v2 to be in line with the title. More dataset, and more quantitative analyses per scan angle (since this is one of the claimed improvement). Also, how well does this approach achieve w.r.t to more classical approaches considered on thermal infrared sensors (e.g. Landsat, METEOSAT, etc...)? Moreover, I don't understand why Section 5 is here reported as an appendix. Some claims are written in the conclusion section, but were never reported before. I had to find out that some additional works were done in this appendix which is not obvious. Also what about the impact of the sea temperature surface? And H2o profile? Does it play a rule somewhere?

In Section 4 we added the results of direct comparison of AIRWAVEv1 and AIRWAVEv2 adding in table 2 the results of AIRWAVEv1 validation.

About the analysis per scan angles: as shown in Fig.0 above, the variations with scan angles depend also on season. AIRWAVEv2 approach is to select the parameters from look up tables through a multivariate interpolation on a 3-dimensional grid (trilinear). Decouple the effects of latitude and scan angle variation is thus not straightforward.

Regarding the validation with additional datasets, this is behind the scope of this paper. Actually, we present here the validation with both satellite data and radiosondes. The intrinsic nature of these two datasets and the AIRWAVE one are very different and independent to guarantee a good benchmark for validation. Furthermore, the scopes of this paper are to present the AIRWAVEv2 dataset, compare its performances with respect to AIRWAVEv1 and describe the new algorithm, as described in the title.

We prefer to keep Section 5 in the Appendix to ease the readability of the paper. Following the reviewer's suggestions, we add in the revised version of the paper an estimate of the impact of additional components such as surface temperature and H2O profile.

In particular, we change the H2O profile varying randomly of 5% H2O at each level and we changed the SST randomly of 3K. When performing this test, we keep the value of the temperature of the last layer equal to the SST (no contrast). We report the results for water vapor in a new column in Table 3, while we prefer to substitute the results of the impact of varying temperature profile alone with the ones obtained when varying both SST and T profile (with no contrast between SST and temperature in the last layer). Consequently, we change the title of section 5.2 "Atmospheric temperature profile" in "Atmospheric temperature and water vapor profiles and SSTs"

In this section we rewrite the section in this way:

"One of the main error sources that can affect the AIRWAVE TCWV retrieval is the assumption of a fixed temperature profile. Actually, the atmospheric opacity is closely linked to the atmospheric density and thus to the atmospheric temperature. To estimate the impact of temperature on the retrieved TCWV, twenty different temperature profiles were randomly perturbed by ±3 K on a 1 km equi-spaced altitude grid. In order to account also for possible changes related to SST variation we change the SST accordingly to the value of the temperature in the lowest layer. Then, simulated BTs were produced with the RTM and were used to perform the TCWV retrievals for the three instruments in equatorial, mid-latitude and polar July conditions for the north hemisphere and the results were compared with respect to the unperturbed case. In the third column of Table 3 we summarise the findings of this analysis for each of the three instruments, reporting the STD of the difference both in absolute and in percentage values. The impact of these perturbations is of the order of 6% and is higher in the equatorial and mid- latitude regions and lower at the poles. Indeed, in the equatorial region, due to the higher water vapour content, the atmosphere is more opaque than at the poles so that temperature variations have a larger impact on the retrieved TCWV. These tests were also performed varying the atmospheric profile alone. Very similar but slightly higher errors are found in these cases.

Another relevant error sources can be due to differences in water vapor profile shape. To evaluate this error, we varied the water vapor profile randomly up to 5% at each atmospheric level. At maximum the impact is of 1% in equatorial case (atmospheric opacity, see the fourth column in Table 3)."

25. Table 1: What are the considered zenith angles considered here for the nadir and forward views?

They are the sub-satellite ones (0 and 55 degrees). We added this information in the Table 1 caption.

26. Table 3: Please check and correct the units. Wind speed and temperature profile cannot be in kg/m2.

In Table 3 we did not report the values of temperature and wind but their impact on the TCWV, for this reason it is in kg/m2.

27. Apart of illustrations, what are the purposes of Figs1-4? I don't see any additional validations with them? Seem they take space with a lot of redundant information. I would advise to group them in 1 single plot with 4 panels. So then you can add more validations that would be kore relevant for this paper.

These figures have no validation purpose. They show the climatology of the TCWV obtained from the AIRWAVEv2 dataset. Following the reviewer's suggestion, we grouped figures 1-4 in two figures, one with TCWV and STD maps for the four months and one for zonal and meridional means.

28. Fig5: What are the differences between "mean" (left panel) and "bias" (right panel)?

None, in the revised version of the paper we used "BIAS" in both.

29. Figs5-6-7: Please precise the time period associated with all these figures. And the considered areas for Fig.5.

These figures cover the entire (A)ATSR period from 1991 to 2012. We added this information in the revised version of the paper in figures caption.

**Proposed additional bibliography**

Zhao-Liang Li, Li Jia, Zhongbo Su, Zhengming Wan & Renhua Zhang (2003) A new approach for retrieving precipitable water from ATSR2 split-window channel data over land area, International Journal of Remote Sensing, 24:24, 5095-5117, DOI: 10.1080/0143116031000096014

J. A. Sobrino & M. Romaguera (2008) Water-vapour retrieval from Meteosat 8/SEVIRI observations, International Journal of Remote Sensing, 29:3, 741-754, DOI: 10.1080/01431160701311267

Lindstrot, R., Preusker, R., Diedrich, H., Doppler, L., Bennartz, R., and Fischer, J.: 1D-Var retrieval of daytime total columnar water vapour from MERIS measurements, Atmos. Meas. Tech., 5, 631-646, https://doi.org/10.5194/amt-5-631-2012, 2012.

We added Lindstrot and Li.

---

## Author Response (AR2)

First of all we gratefully thank the reviewer for the suggestions. Our replies are reported in blue below each reviewer's comment.

Dear Authors,

I would like to thank you very much for addressing all my comments and questions in my previous review. Your first version of the manuscript already included a lot of works. But the overall high amount of precise answers and solid clarifications satisfy my raised comments. In particular, I very much appreciate the efforts of having added the Figure 1, merged the Panels of Figure 2, added the discussion of Airwaves 1 vs. 2 + Table 2, the clarification of the equations in Sect. 2.2, and all the quantitative numbers that allow to better understand the quality and sensitivity of your retrievals (overall and per error source).

I recommend the publication in AMT. I have a few remaining minor requests:

- Perhaps it should be directly obvious, but I don't directly see the transition from Eq 12 to Eq 13. Could you please explicit it? Furthermore, what is the meaning of "pseudo-column"? A column integrated along the average light path (by opposition of vertical atmospheric layers?

In Casadio et al., 2016 Φ is named as "water vapor pseudo-column" because the TCWV can be obtained as the sum of this term plus the ratio between the G parameter and the delta sigma one.
The transition from equation 12 to 13 is reported in details in Casadio et al., 2016 in section 3.3. Since this derivation in details is quite long we prefer to avoid repetition of a large part of an already published work. We prefer to summarize as follows:
"the TCWV can be estimated through the knowledge of G. Actually, for single view geometry the presence of the G term can affect the accurate determination of TCWV. Thus, the G variability suggests that it will be desirable to avoid this term in TCWV derivation. In AIRWAVE this is performed exploiting the dual view capability of the ATSR instruments and by assuming a perfect collocation between NAD and FWD measurements:"
We replaced "while the TCWV given by the combined use of nadir and forward pseudo column is: " with these sentences between equation 12 and 13.

- I somehow feel your abstract (and conclusion) underestimates a little bit the impact of your study. I don't think AIRWAVES "only" improves thanks to a higher performance. The improvements are also: a better algorithm implementation by explicitly taking into account the geometry and latitude dependence for each pixel retrieval (a post-correction may create some strange spatial artefacts, hard to quantify in terms of performances), and a better uncertainty estimation of your overall retrievals with a large sensitivity analyses to various parameters. This last one is, for someone like me, important as it may better recommend me to use this dataset (and its

overall error bar) for future applications. You may want to stress that. You also (now) made a direct comparison with previous Airwaves v1.

In the revised version of the paper we add these considerations in the conclusions session.
In line 317 we add: "These improvements are due to two factors. First the AIRWAVEv2 retrieval parameters now account for the atmospheric variability. Secondly the implementation of the new algorithm explicitly takes into account the geometry and latitude dependence of each pixel, allowing to overcome possible artefacts due to approximations and a posteriori corrections."
While in line 326 we add:"The error quantification given in this work allows the users to get a better insight of the AIRWAVEv2 TCWV dataset and of its related quality".

- I found your Figure 0 in your reply informative. Why not adding it in your new manuscript?

In the revised version of the paper we add the figure in the manuscript. In addition we add this part of text in line 211:

[revised manuscript text omitted]